DOI: 10.1038/s41467-017-02358-7　　**OPEN**

# TNFα blockade overcomes resistance to anti-PD-1 in experimental melanoma

Florie Bertrand[1,2], Anne Montfort[1,2], Elie Marcheteau[1,2,3,4], Caroline Imbert[1,2,3,4], Julia Gilhodes[5], Thomas Filleron[5], Philippe Rochaix[5], Nathalie Andrieu-Abadie[1,2], Thierry Levade[1,2,3,4,6], Nicolas Meyer[1,3,4,7], Céline Colacios[1,2,3,4] & Bruno Ségui[1,2,3,4]

Antibodies against programmed cell death-1 (PD-1) have considerably changed the treatment for melanoma. However, many patients do not display therapeutic response or eventually relapse. Moreover, patients treated with anti-PD-1 develop immune-related adverse events that can be cured with anti-tumor necrosis factor α (TNF) antibodies. Whether anti-TNF antibodies affect the anti-cancer immune response remains unknown. Our recent work has highlighted that TNFR1-dependent TNF signalling impairs the accumulation of CD8+ tumor-infiltrating T lymphocytes (CD8+ TILs) in mouse melanoma. Herein, our results indicate that TNF or TNFR1 blockade synergizes with anti-PD-1 on anti-cancer immune responses towards solid cancers. Mechanistically, TNF blockade prevents anti-PD-1-induced TIL cell death as well as PD-L1 and TIM-3 expression. TNF expression positively correlates with expression of PD-L1 and TIM-3 in human melanoma specimens. This study provides a strong rationale to develop a combination therapy based on the use of anti-PD-1 and anti-TNF in cancer patients.

[1] INSERM UMR 1037, CRCT, 31037 Toulouse, France. [2] Equipe Labellisée Ligue Contre Le Cancer, 31037 Toulouse, France. [3] Université Toulouse III - Paul Sabatier, 31062 Toulouse, France. [4] Université Fédérale de Toulouse Midi-Pyrénées, 41 Allée Jules Guesde, 31000 Toulouse, France. [5] Institut Universitaire du Cancer, 31059 Toulouse, France. [6] Laboratoire de Biochimie, Institut Fédératif de Biologie, CHU Purpan, 31059 Toulouse, France. [7] Institut Universitaire du Cancer, Toulouse, Hôpital Larrey et Oncopôle, 31059 Toulouse, France. Florie Bertrand and Anne Montfort contributed equally to this work. Céline Colacios and Bruno Ségui jointly supervised this work. Correspondence and requests for materials should be addressed to B.Ség. (email: bruno.segui@inserm.fr)

Tumor necrosis factor α (TNF) plays a dual role in oncoimmunology[1–3], either acting as an anti-cancer factor[4,5], or behaving as an immunosuppressive cytokine[6–11]. Whereas TNF was first identified as a cytotoxic soluble factor[4], mounting evidence indicates that TNF facilitates the accumulation and/or biological activity of regulatory T lymphocytes (Tregs)[12], regulatory B lymphocytes (Bregs)[8] as well as myeloid-derived suppressor cells (MDSC)[9], which are key negative modulators of the immune response. Moreover, TNF triggers activation-induced cell death (AICD) of CD8+ T cells[13] and impairs the tumor infiltration by CD8+ T lymphocytes[1,2,14]. CD4 + tumor-infiltrating lymphocytes (TILs) in melanoma produce TNF, which inhibits cytotoxic CD8+ T-cell responses[15]. In an adoptive CD8+ T-cell transfer protocol, TNF elicited melanoma dedifferentiation, promoting immune escape and melanoma relapse[16]. More recently, TNF has been shown to enhance the expression of programmed cell death ligand 1 (PD-L1) in cancer cells[17], including melanoma[15], triggering immunosuppression.

Emerging immunotherapies that target immune checkpoints, like anti-PD-1 (programmed cell death-1), have radically changed our strategy to fight melanoma[18]. However, 60–67% of patients do not respond to anti-PD-1[19,20], and a significant proportion of responders experience tumor relapse within two years after treatment induction[19–21]. One of the documented cause of this relapse was shown to be the expression of secondary immune checkpoints such as T-cell Immunoglobulin and Mucin domain-containing protein 3 (TIM-3)[22]. Anti-PD-1 treatment is associated with increased TNF gene expression in melanoma samples from metastatic melanoma patients[23]. In this context, the contribution of TNF to the anti-melanoma immune response is unknown. In addition, responders develop severe immune-related adverse events (irAEs) with a frequency close to 12%[19,20]. Whereas anti-TNF antibodies are used to cure irAEs associated with immunotherapies (such as anti-CTLA-4, anti-PD-1 or a combination of both)[24], whether TNF blockade affects the anti-melanoma immune response triggered by immunotherapies remains to be determined.

Herein, we have evaluated the consequences of blocking the TNFR1-dependent TNF signalling on anti-PD-1 efficacy in experimental cancer. We provide evidence for the first time that TNF impairs the response to anti-PD-1 not only in mouse melanoma but also in two other experimental cancers (i.e., lung and breast cancer). Mechanistically, TNF induced upon anti-PD-1 treatment triggers (i) the expression of both PD-L1 and TIM-3 on CD4+ and CD8+ TILs and (ii) AICD in CD8+ TILs, which co-express PD-1 and TIM-3. Consequently, TNF blockade/deficiency prevents PD-L1 and TIM-3 expression as well as AICD in CD8+ TILs, overcoming the resistance to anti-PD-1 in experimental melanoma.

## Results

**TNF/TNFR1 deficiency potentiates anti-PD-1 treatment efficacy.** We investigated the impact of TNF deficiency on the therapeutic effect of anti-PD-1 on melanoma development. To this end, wild-type (WT) and TNF-deficient mice were bilaterally injected with $3.10^5$ melanoma cells. When tumors became detectable (i.e., palpable, at day 6), mice received the first out of three injections of anti-PD-1 blocking antibody or isotype control. In agreement with previous observations[14], B16K1 tumor growth (Figs. 1a and b) and animal death (Fig. 1c) were significantly delayed in TNF-deficient mice as compared to their WT counterparts. In WT animals, anti-PD-1 injections significantly delayed the B16K1 tumor growth. However, most of the tumors relapsed and all mice died within 50–60 days post-tumor cell injection (Figs. 1a and c). Remarkably, in TNF-deficient mice,

anti-PD-1 therapy led to the rejection of 90% of tumors (Figs. 1a and b). Moreover, more than 80% of these mice survived up to 120 days (Fig. 1c). Similar results were obtained when anti-PD-1 was administered to WT and TNF-deficient mice bearing larger tumors (about 25 mm$^3$ at day 6) (Figs. 1d–f). Of interest, all TNFR1-deficient mice (Supplementary Fig. 1) as well as WT mice injected with an anti-TNFR1 blocking antibody (Supplementary Fig. 2) rejected melanoma cells upon anti-PD-1 therapy. Two months after the first B16K1 melanoma cell inoculation, B16K1 melanoma cell re-injection in the surviving TNF- and TNFR1-deficient mice did not compromise overall survival, demonstrating that animals were totally vaccinated towards B16K1 melanoma cells (Fig. 1c and Supplementary Fig. 1c).

To extend our observations to another cancer model, we tested the effect of anti-PD-1 blocking antibodies in WT, TNF- and TNFR1-deficient mice injected with Lewis lung carcinoma (LLC) cells, which expressed MHC-I at low levels (Supplementary Fig. 3a). Neither TNF nor TNFR1 deficiency impaired LLC growth. Moreover, in WT mice, anti-PD-1 had no effect on tumor growth indicating that LLC were fully resistant to this therapy under our experimental conditions (Supplementary Fig. 3b and c). Of interest, TNF or TNFR1 deficiency partially overcame the resistance of LLC to anti-PD-1 (Supplementary Fig. 3b and c). Thus, synergism of TNF/TNFR1 deficiency and anti-PD-1 is likely not restricted to melanoma but applies to other cancers, such as lung carcinoma, which can benefit from anti-PD-1 according to phase 2 and phase 3 clinical trials[25,26]. Collectively, our data indicate that the TNFR1-dependent TNF signalling impairs the therapeutic benefit of PD-1 blockade.

**TNF deficiency reduces TIL cell death upon anti-PD-1 therapy.** To get insights into the molecular and cellular mechanisms involved in the beneficial effect of TNF blockade in anti-PD-1 therapy, we bilaterally grafted one million B16K1 cells in WT and TNF-deficient mice followed by a single anti-PD-1 injection seven days later. At day 10, tumor weights were significantly reduced in TNF-deficient mice as compared to WT mice, and this phenomenon was further amplified upon anti-PD-1 injection (Fig. 2a). Interestingly, whereas a single anti-PD-1 injection failed to promote tumor rejection in WT mice, it triggered total rejection of more than 30% of tumors in TNF-deficient mice (Fig. 2a).

We next investigated the effect of anti-PD-1 on TIL accumulation. Analysis of leukocyte tumor content by flow cytometry indicated that anti-PD-1 injection in WT mice induced a small, yet significant, increase in CD45+ and Thy1+ T cells (Fig. 2b), including both CD8+ and, albeit to a lesser extent, CD4 + TILs (Fig. 2c). In line with previous findings[14], CD8+ TILs were dramatically increased in TNF-deficient mice, with no further elevation upon anti-PD-1 treatment (Fig. 2c).

To evaluate whether TNF deficiency impacts on TIL proliferation, we monitored Ki67 expression, a proliferation marker. The proportions of Ki67+ CD8+ TILs and Ki67+ CD4+ TILs were similar in WT and TNF-deficient mice treated with anti-PD-1 (Supplementary Fig. 4). Thus, TNF does not seem to be a major regulator of TIL proliferation under anti-PD-1 therapy. We next monitored TIL cell death by evaluating the increase in plasma membrane permeability (Supplementary Fig. 5). Importantly, in anti-PD-1-treated animals, TNF deficiency significantly reduced the death of both CD8+ and CD4+ TILs (Fig. 2d). Of note, TNF deficiency did not impair chemokine mRNA levels in tumors upon anti-PD-1 therapy (Supplementary Fig. 6). Collectively, our data indicate that TIL accumulation in TNF-deficient mice resulted from a decreased TIL cell death rather than an increase in T cell proliferation or chemotaxis under anti-PD-1 therapy.

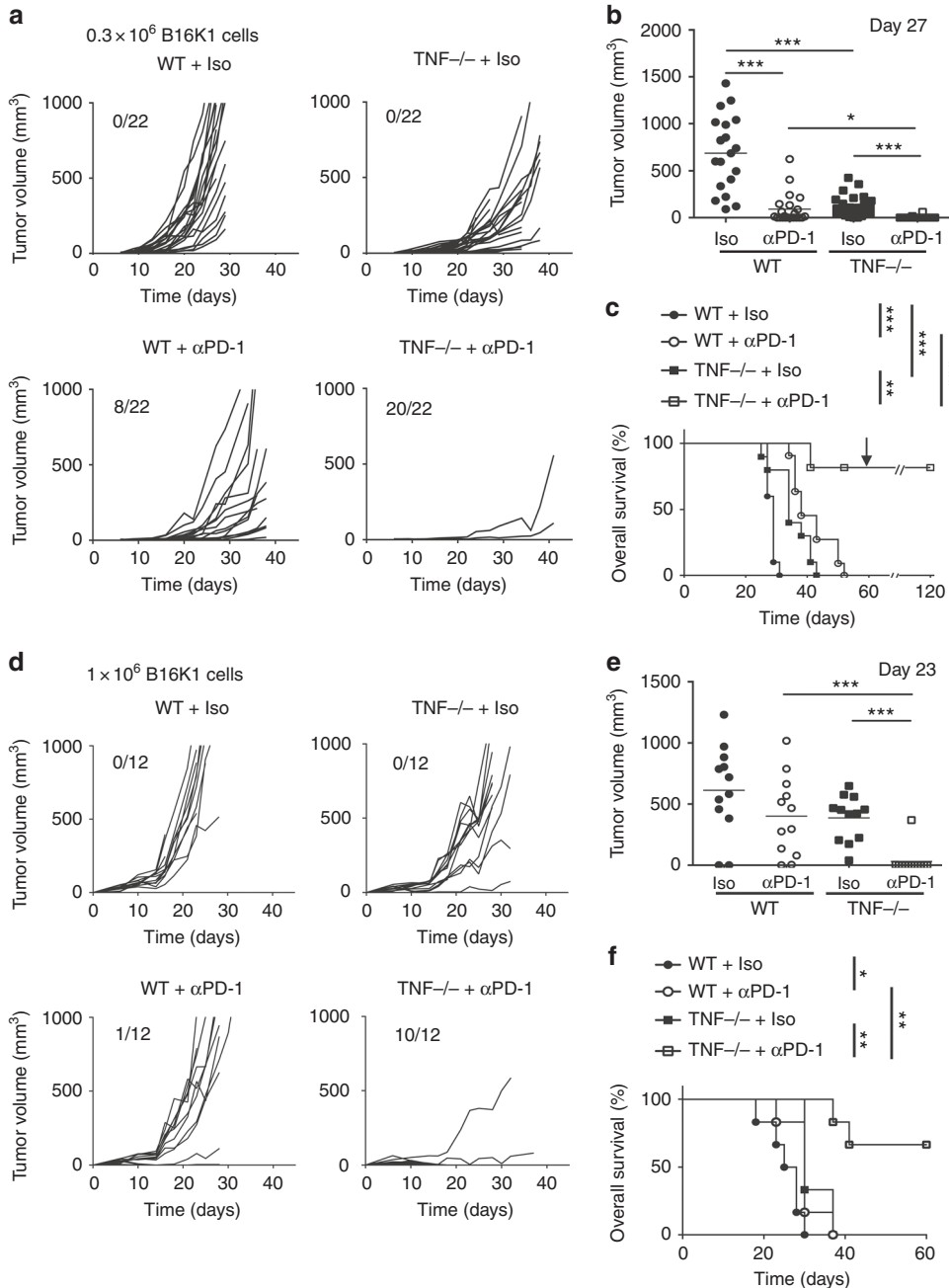

**Fig. 1** TNF deficiency enhances anti-PD-1 response in a mouse melanoma model. C57BL/6 wild-type (WT) and TNF-deficient (TNF−/−) mice were intradermally and bilaterally grafted with $3 \times 10^5$ (**a–c**) ($n = 11$ mice per group) or $1 \times 10^6$ (**d–f**) ($n = 6$ mice per group). B16K1 melanoma cells prior to intraperitoneal injection of anti-PD-1 antibodies (αPD-1, 10 mg kg$^{-1}$) or a relevant isotype control (Iso, 10 mg kg$^{-1}$) at days 6, 10 and 13. **a** and **d** Tumor volumes were determined with a calliper. Individual curves are depicted for each tumor. Numbers indicate complete tumor regression out of total tumors. Data are from three independent experiments. **b** and **e** Tumor volumes determined at the indicated days for individual tumors are depicted. Bars represent mean values ± s.e.m. (Mann–Whitney U test: *$p < 0.05$; **$p < 0.01$; ***$p < 0.001$). **c** and **f** Cumulative survival curves (Logrank test: *$p < 0.05$; **$p < 0.01$; ***$p < 0.001$). At day 60, surviving mice were challenged with a second B16K1 injection as indicated by the arrow; mice did not develop tumors and survived (**c**)

**TNF deficiency does not alter CD8+ TIL effector functions**. In WT mice, anti-PD-1 injection resulted in increased levels of both TNF and IFN-γ transcripts in tumors (Supplementary Fig. 7a and b). Interestingly, in tumors from TNF-deficient mice, IFN-γ transcripts were increased under basal conditions and further enhanced upon anti-PD-1 treatment (Supplementary Fig. 7b). In tumors from CD8-deficient mice, TNF and IFN-γ mRNA levels remained low under both control and anti-PD-1 conditions, demonstrating that CD8+ T cells are required for potent

production of TNF and IFN-γ in this melanoma model (Supplementary Fig. 7a and b).

Whereas the proportion of IFN-γ+ CD8+ TILs was slightly, yet significantly, reduced by TNF loss (Supplementary Fig. 7c), lymphocytes from anti-PD-1-treated TNF-deficient mice likely remained cytotoxic as demonstrated by the conserved proportion of granzyme B+ CD8+ TILs in these animals compared to the WT ones (Supplementary Fig. 7d). Of note, TNF deficiency led to a significant increase in the proportion of granzyme B+ CD8+

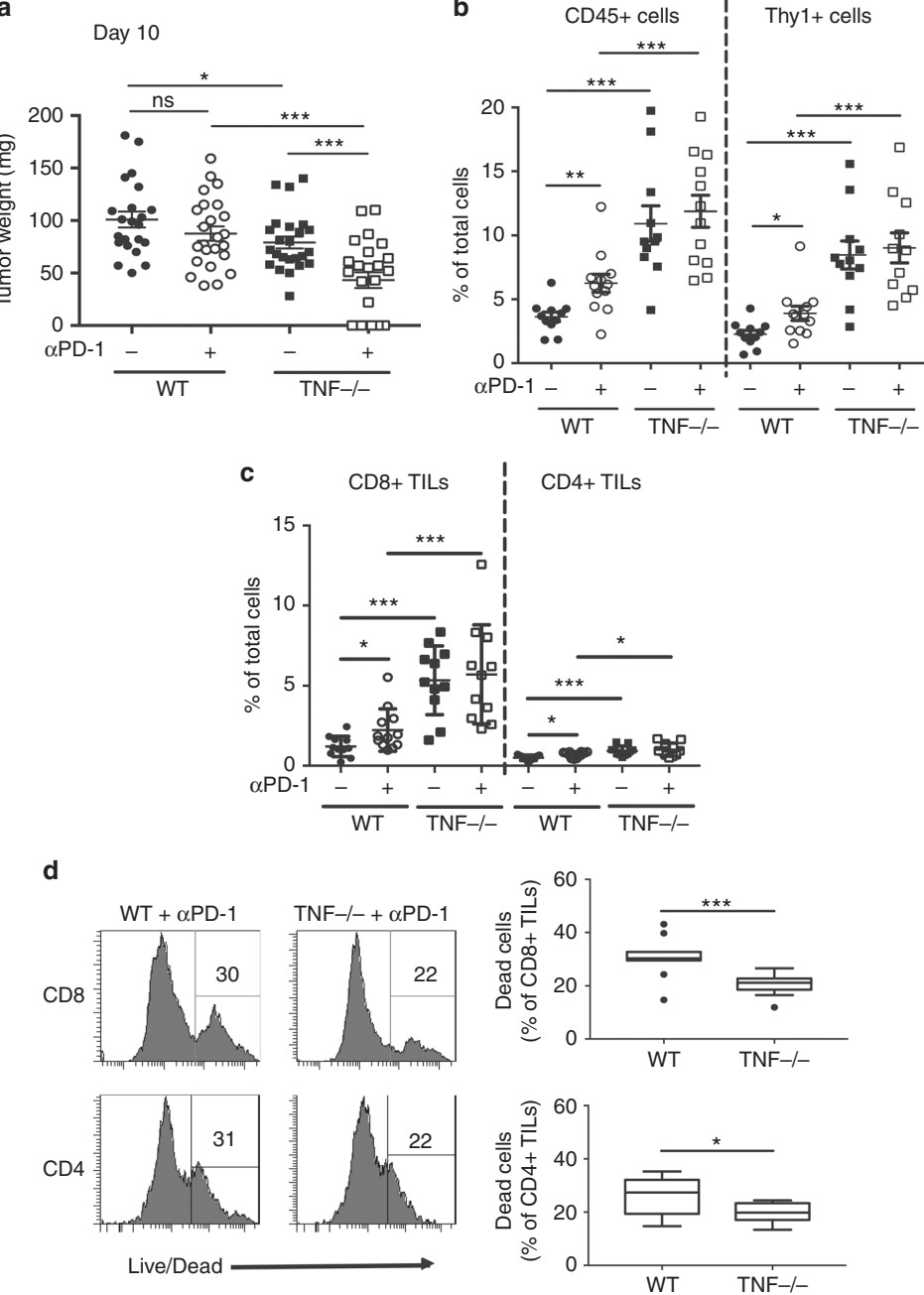

**Fig. 2** Immune cell infiltration of tumors from anti-PD-1 treated wild-type and TNF-deficient mice with established melanoma. **a–c**, C57BL/6 wild-type (WT) and TNF-deficient (TNF−/−) mice were intradermally and bilaterally grafted with $1 \times 10^6$ B16K1 melanoma cells prior to intraperitoneal injection of anti-PD-1 antibodies (αPD-1, 10 mg kg$^{-1}$) or vehicle (PBS) at day 7. **a** At day 10, mice were sacrificed and tumors were weighed. Data are means ± s.e.m. of at least 22 tumors per group from two independent experiments (Student's t-test: *p < 0.05; ***p < 0.001). **b** and **c** TIL content was analysed by flow cytometry. The proportion of CD45+ and Thy1+ cells (**b**) CD4+ and CD8+ TILs (**c**) among total cells was determined. Data are means ± s.e.m. of at least 11 tumors per group from two independent experiments (Mann–Whitney U test: *p < 0.05; **p < 0.01; ***p < 0.001). **d** Proportion of dead CD8+ and CD4+ TILs from WT and TNF-deficient mice treated with anti-PD-1. Left panels: representative stainings. Right panels: values measured in 12 tumors per group from two independent experiments are represented as Tukey boxes (Student's t-test: *p < 0.05; ***p < 0.001)

TILs independently of anti-PD-1 treatment (Supplementary Fig. 7d). No difference in the proportion of IFN-γ+ CD4+ TILs between anti-PD-1-treated WT and TNF-deficient mice was observed (data not shown).

TNF is known to promote PD-L1 expression in solid cancers[17], including melanoma[15]. We thus analyzed PD-L1 and PD-L2 levels under our experimental conditions. In WT mice, treatment with anti-PD-1 selectively enhanced PD-L1, but not PD-L2, expression by both CD4+ and CD8+ TILs (Supplementary Fig. 8a

and b). TNF deficiency totally abrogated PD-L1 up-regulation on TILs upon anti-PD-1 therapy (Supplementary Fig. 8a and b). We also monitored the tumor content of dendritic cells (DCs), which play a key role in the priming of T cells. Tumor-infiltrating DCs were slightly, yet significantly, increased in WT mice treated with anti-PD-1 as well as in TNF-deficient mice with or without anti-PD-1 injection (Supplementary Fig. 8c). Moreover, the expression of PD-L1 and, albeit to a lesser extent, PD-L2 was reduced on DCs in TNF-deficient mice under control and anti-PD-1

conditions (Supplementary Fig. 8d–f). Thus, TNF deficiency favors DC accumulation in tumors, while reducing, but not abrogating, the expression of the PD-1 ligands.

**TNF augments αPD-1-stimulated TIM-3 up-regulation on TILs.** TIM-3 inhibits Th1 responses and triggers peripheral tolerance[27]. This newly recognized key immune checkpoint impairs the antigen-specific CD8+ T cell response in melanoma[28] and plays a major role in the immune escape mechanisms. This was associated with resistance to anti-PD-1 both in mice and patients[22]. The mean fluorescence intensity (MFI) of TIM-3 staining, which reflects its expression on TILs, was significantly increased on CD8+ and CD4+ TILs following anti-PD-1 administration in WT mice (Supplementary Fig. 9a and b). In sharp contrast, the TIM-3 up-regulation was totally abolished in TNF-deficient mice (Supplementary Fig. 9a and b), suggesting that the anti-PD-1-triggered TIM-3 expression occurred in a TNF-dependent manner. Of note, TNF-deficiency did not impair the expression of TIGIT, LAG3 and CTLA-4 on CD8+ TILs (Supplementary Fig. 10), indicating that TNF is not a potent inducer of these immune checkpoint molecules under our experimental conditions.

To evaluate the putative functional impact of TIM-3 expression, we monitored both IFN-γ and granzyme B expression by CD8+ TILs 3 days after a single anti-PD-1 injection and 10 days after tumor cell inoculation (Supplementary Fig. 11). Analysis at later time points could not be performed because of complete tumor regression in most of the TNF-deficient mice. The proportion of CD8+ TILs expressing IFN-γ and granzyme B was higher in TIM-3 cells in each experimental condition (Supplementary Fig 11a and b), indicating that under our setting TIM-3 represents an early activation marker rather than an exhaustion marker at this stage. Moreover, the proportion of IFN-γ+ and granzyme B+ TIM-3+ CD8+ TILs was slightly, yet significantly, reduced in anti-PD-1-treated TNF-deficient mice as compared to WT mice. Strikingly, anti-PD-1-triggered CD8+ TIL AICD occurred in TIM-3+ cells only and this was totally abrogated by TNF deficiency (Supplementary Fig. 11c).

We next analyzed the co-expression of TIM-3 and PD-1 on CD8+ TILs in each experimental condition (Fig. 3a). Whereas the proportion of PD-1+ CD8+ TILs remained unchanged by anti-PD-1 therapy and TNF-deficiency, the proportion of TIM-3+ CD8+ TILs significantly increased upon anti-PD-1 in WT but not TNF-deficient mice (Fig. 3b). Consequently, the proportion of CD8+ TILs expressing simultaneously TIM-3 and PD-1 was lower in TNF-deficient mice (Fig. 3c). Most importantly, this specific cell population was the only one significantly affected by cell death following anti-PD-1 treatment, an effect abrogated by TNF deficiency (Figs. 3c and d).

To evaluate whether TNF can induce TIM-3 expression, we purified and activated naive murine CD8 + T cells prior to incubation with exogenous and sub-toxic doses of murine TNF. TNF elicited an increase of both the percentage of TIM-3+ CD8+ T cells and the MFI of TIM-3 staining (Fig. 4a). This was suppressed in TNFR1-deficient CD8+ T cells, indicating that TNF induced TIM-3 expression in a TNFR1-dependent manner (Fig. 4a). Moreover, TNF triggered a modest TIM-3 up-regulation on non-activated CD8+ T cells (data not shown). Whereas TNF potently upregulated TIM-3 expression on activated CD8+ T cells, it had no or minimal effect on TIGIT and LAG3 expression (Supplementary Fig. 12a).

We next investigated the molecular mechanisms involved in TNF-induced TIM-3 expression on activated CD8+ T cells. We focused our analysis on the signaling pathways, i.e., MEK, p38 MAPK and PI3K, which are activated by TNF[29] and lead to TIM-

3 expression[30,31]. Pharmacological inhibition of PI3K and MEK, but not p38 MAPK, significantly reduced basal TIM-3 expression. Interestingly, only the p38 MAPK inhibitor totally prevented TNF-induced TIM-3 expression on activated CD8+ T cells (Supplementary Fig. 12b). Thus, p38 MAPK likely plays a critical role in TIM-3 up-regulation on CD8+ T cells in response to TNF.

Of interest, TNF also stimulated the expression of TIM-3 on naive and activated human CD8+ T cells purified from the peripheral blood of healthy donors (Supplementary Fig. 12c). We next sought to evaluate the effect of exogenous TNF on CD8+ TILs purified from human melanoma samples under basal conditions (i.e., IL-2 alone) or after autologous re-stimulation with melanoma cells for 48 h. TNF potently stimulated TIM-3 expression on CD8+ TILs and this was enhanced upon autologous co-culture with melanoma cells (Figs. 4b and c). The TNF effect on TIM-3 expression was dose-dependent in a distinct autologous co-culture experiment using TILs from another metastatic melanoma patient (Supplementary Fig. 12d).

Collectively, our data demonstrate for the first time that TNF is a potent inducer of TIM-3 expression on CD8+ TILs both in experimental mouse melanoma and in vitro using TILs isolated from metastatic melanoma patients.

**Anti-TNF antibodies potentiate anti-PD-1 efficacy.** To evaluate the therapeutic relevance of the above observations, we tested the combination of anti-TNF and anti-PD-1 blocking antibodies in mouse melanoma (Figs. 5a and b). As compared to untreated mice, anti-TNF alone slightly inhibited B16K1 melanoma growth (Fig. 5a). Combining anti-PD-1 and anti-TNF led to a total regression of 75% of the tumors (Fig. 5a) and led to the survival of 75% of mice as compared to less than 20% in the group of animals injected with anti-PD-1 alone (Fig. 5b). Surviving mice were totally vaccinated against B16K1 (Fig. 5b). Of note, anti-TNF also synergised with anti-PD-1 in a breast cancer model based on orthotopic graft of 4T1 cells in the mammary fat pad of Balb/c mice. Whereas anti-PD-1 or anti-TNF alone failed to impair tumor growth, their combination significantly decreased it (Supplementary Fig. 13).

To further investigate the molecular and cellular mechanisms involved in the therapeutic benefit of combining anti-PD-1 and anti-TNF, we analysed the CD8+ TILs in B16K1 tumors. Anti-TNF augmented the proportion of CD8+ TILs (Fig. 5c) and prevented not only their death (Fig. 5d) but also their up-regulation of TIM-3 (Figs. 6a–c) in response to anti-PD-1 treatment. These findings show that TNF produced upon anti-PD-1 therapy triggers both AICD of CD8+ TILs and TIM-3 up-regulation on TILs. We next compared the efficacy of early anti-PD-1 therapy associated or not with anti-TNF and/or anti-TIM-3 (Fig. 6d). The combination of anti-TIM-3 and anti-PD-1 slightly, yet not significantly, attenuated tumor growth as compared to anti-PD-1 alone, whereas combining anti-TNF and anti-PD-1 proved to be more potent at reducing and slowing down tumor growth (Fig. 6d). We performed another experiment with the first antibody injection delayed to day 13. Under those experimental conditions, anti-PD-1, anti-TIM-3 or anti-TNF alone failed to impair tumor growth (Fig. 6e and data not shown). Co-administrations of anti-PD-1 with anti-TIM-3 or anti-TNF were equally effective at reducing tumor growth as compared to anti-PD-1 alone or vehicle (Fig. 6e). Finally, co-injecting anti-TIM-3 together with anti-TNF and anti-PD-1 did not further improve the therapeutic effect (Figs. 6d and e). Similar findings were observed in TNF-deficient mice, in which anti-TIM-3 injection did not enhance anti-PD-1 therapeutic effects (data not shown). Altogether, our observations indicate that the therapeutic benefit

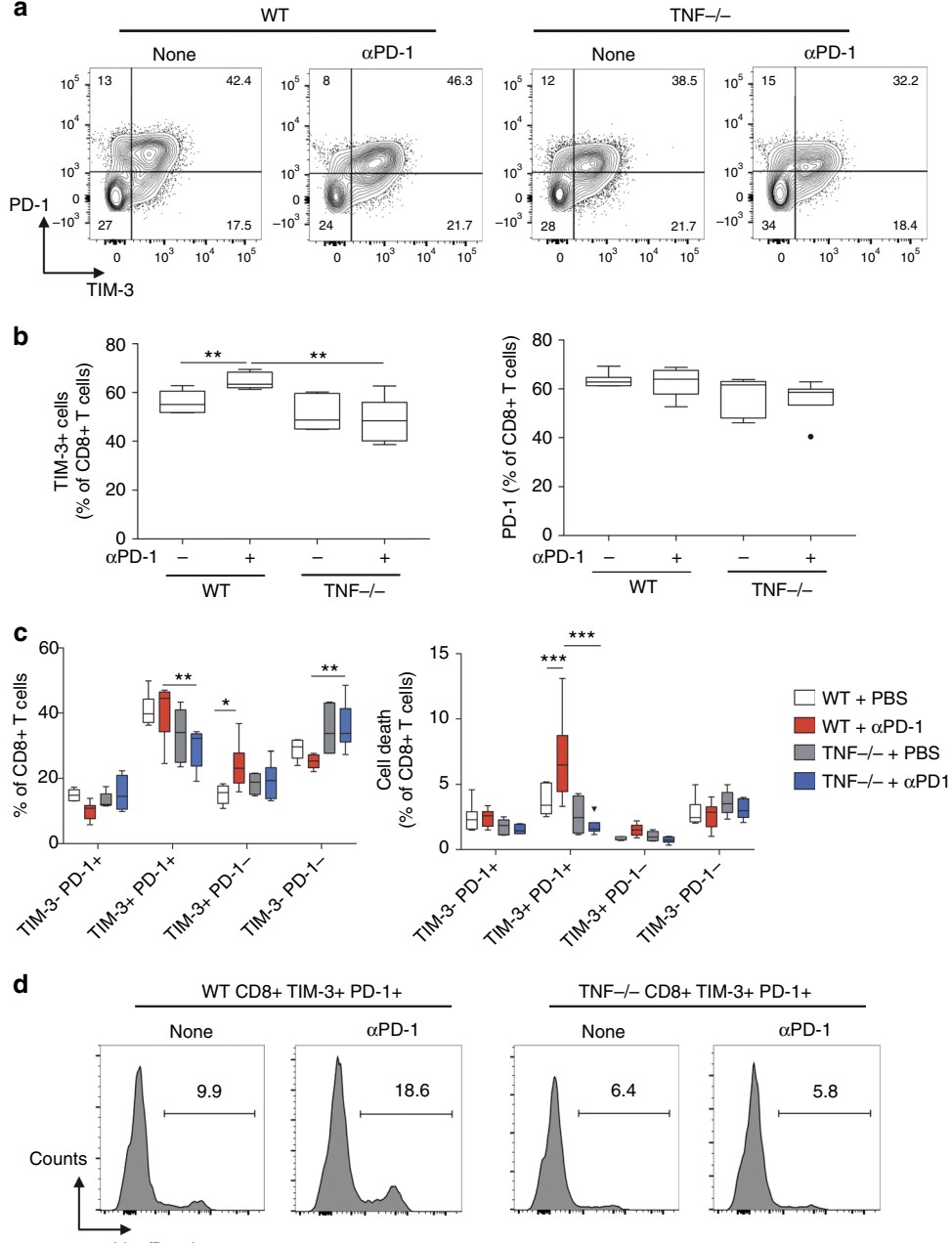

**Fig. 3** Anti-PD-1 triggers TIM-3 expression on CD8+ TILs in a TNF-dependent manner. WT and TNF-deficient mice were injected as described in the legend to Fig. 2. TIM-3 and PD-1 expression on CD8+ TILs was determined by flow cytometry at day 10. **a** Representative staining; values indicate the proportion of cells in the different quadrants. **b** Quantification of the proportion of TIM-3+ (left panel) and PD-1+ (right panel) among CD8+ TILs. **c** Quantification of the proportion of CD8+ TILs expressing or not TIM-3 and PD-1 (left panel), and proportion of cell death among CD8+ T cells of the indicated populations (right panel). Values in 5–6 mice per group from one experiment are represented as Tukey boxes (b: Student's $t$-test: $^{**}p < 0.01$; c: two-way Anova: $^{***}p < 0.001$). **d** Representative histograms of live/dead staining in CD8+ TILs expressing both TIM-3 and PD-1. Values are percentages of dead cells among the TIM-3+ PD-1+ CD8+ TILs

of anti-TNF and anti-PD-1 combination relies, at least in part, on the inhibition of the TIM-3-dependent pathway.

**TNF and immune escape gene signature in human melanoma**. To evaluate how our findings translate to advanced melanoma in patients, we next explored the TCGA melanoma data bank[32] for the expression of a set of genes encoding proteins involved in immune escape and, eventually, resistance to anti-PD-1 therapy. Strikingly, an immune escape gene signature was evidenced in

melanoma samples exhibiting high TNF expression, suggesting that TNF is part of a gene network, which leads to immune suppression in human melanoma (Fig. 7a). In addition, the expression of *TNFA* (encoding TNF) positively and significantly correlated with that of *HAVCR2* (encoding TIM-3), *PDCD1LG1* (encoding PD-L1) and *PDCD1LG2* (encoding PD-L2) (Fig. 7b). Our analysis of another published dataset[23] also revealed a strong positive and significant correlation between *TNFA*, *HAVCR2*, *PDCD1LG1* and *PDCD1LG2* expression in melanoma specimens from patients treated with anti-PD-1 (Fig. 7c).

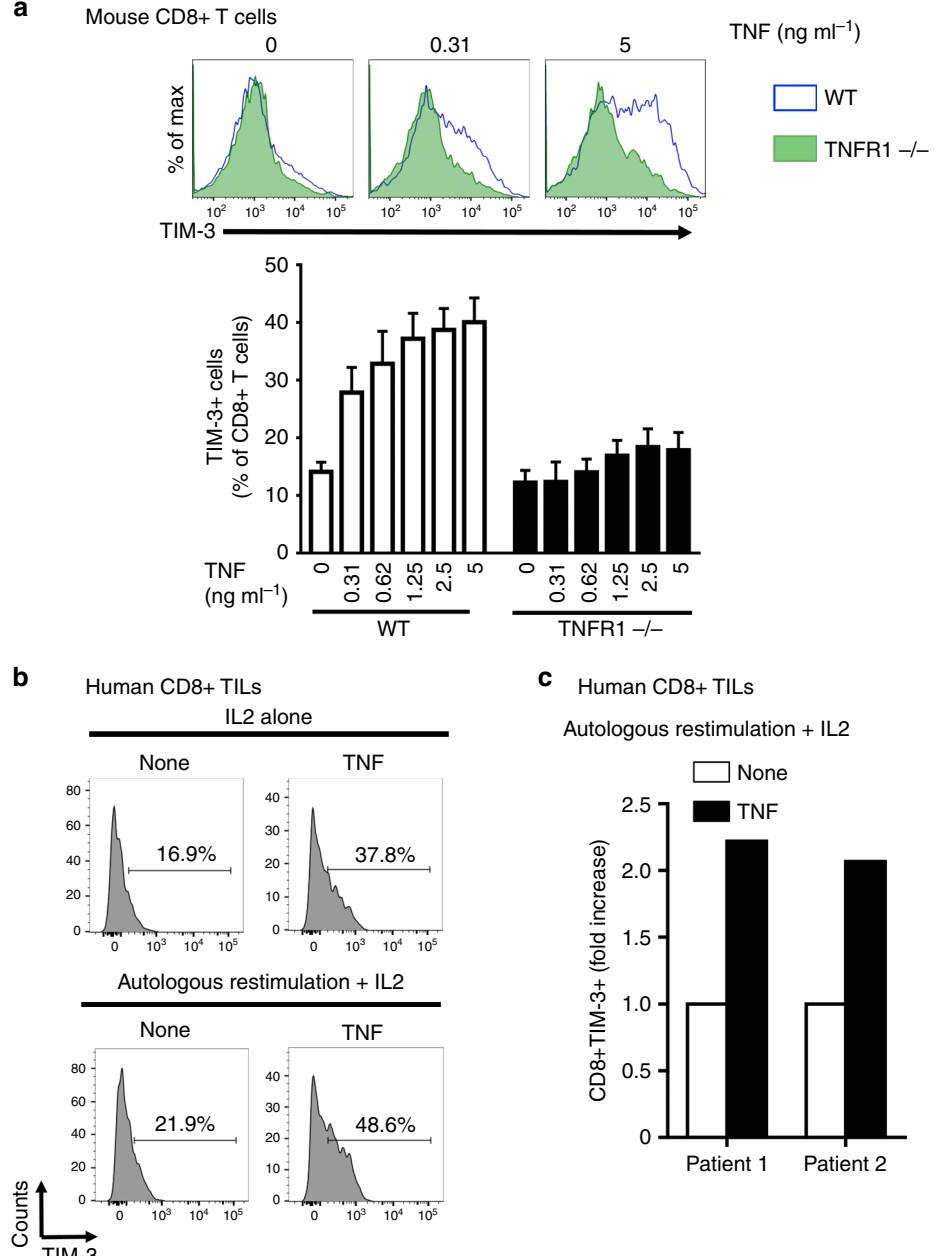

**Fig. 4** TNF induces TIM-3 expression on CD8+ T cells ex vivo. **a** WT or TNFR1-deficient CD8+ T cells were incubated with murine TNF for 2 days. TIM-3 expression was next analysed by flow cytometry. Upper panels: representative histograms. Lower panel: data are means ± s.e.m. of three independent experiments. **b** and **c** TILs from two human metastatic melanoma patients were cultured with or without autologous melanoma cells for two days in the presence 200 U ml$^{-1}$ IL-2 +/−50 ng ml$^{-1}$ human TNF. TIM-3 expression was next analysed by flow cytometry: histograms showing TIM-3 staining on TILs from patient 1 (**b**); bar graph depicting the fold increase in TIM-3 expression on TILs from patients 1 and 2 (**c**)

Thus, along with our preclinical data, those observations on human melanoma suggest that TNF potently induces the expression of PD-L1, PD-L2 and TIM-3 in melanoma upon anti-PD-1 therapy.

## Discussion

One of the limitations to the successful use of anti-PD-1 to treat cancer patients is the requirement for tumors to be pre-infiltrated with immune cells and more specifically TILs[33]. However, a high proportion of patients affected by cancers that are described as "immunogenic" still poorly respond to these therapies. As opposed to the "primary resistance", these resistance mechanisms

are defined as "adaptive immune resistance" and "acquired resistance"[34]. One documented cause for this phenomenon is the increased expression of secondary immune checkpoint molecules such as TIM-3 that dampen the anti-tumor immune response[22]. This study provides the first evidence that TNFR1-dependent TNF signalling constitutes a potent immune escape mechanism in a context of T-cell-inflamed tumor microenvironment, conferring resistance to anti-PD-1.

We show here that TNF blockade leads to an increased content of CD8+ TILs, which is predictive of the response to anti-PD-1 in melanoma patients[33]. This phenomenon was also associated with a reduced expression of PD-L1 on DCs and TILs, in agreement with recent papers demonstrating the role of TNF in PD-L1

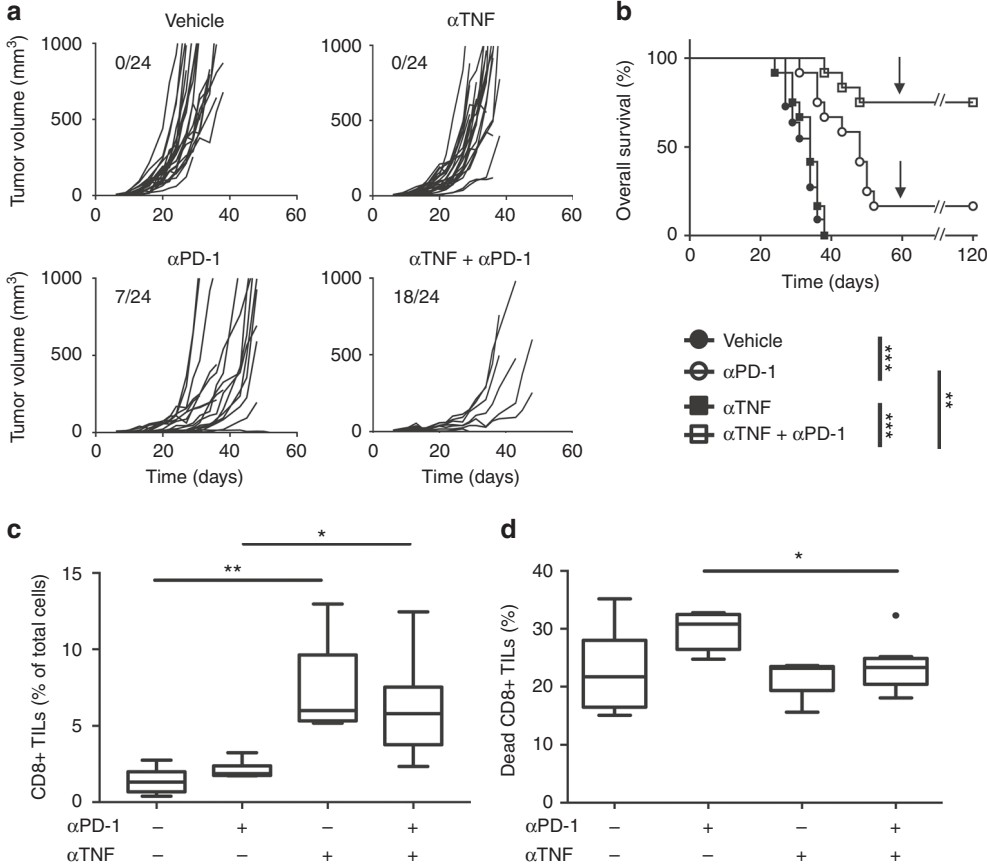

**Fig. 5** Anti-TNF treatment enhances anti-PD-1 response in a mouse melanoma model. C57BL/6 wild-type (WT) mice were intradermally grafted with $3 \times 10^5$ B16K1 melanoma cells followed by intraperitoneal injection of anti-TNF antibodies ($\alpha$TNF, 10 mg kg$^{-1}$) or vehicle (PBS) at days 6, 9, 13 and 16 and/or anti-PD-1 (10 mg kg$^{-1}$) at days 6, 9 and 13 ($n = 12$ mice per group). Data are from two independent experiments. **a** Individual tumor volumes are depicted. Numbers indicate complete tumor regression out of total tumors. **b** Cumulative survival curves. At day 60, (arrow) surviving mice were challenged with a second B16K1 injection; these mice did not develop tumors and survived (Logrank test: **$p < 0.01$; ***$p < 0.001$). **c** and **d** C57BL/6 WT mice were intradermally and bilaterally grafted with $1 \times 10^6$ B16K1 melanoma cells prior to intraperitoneal injection of anti-TNF ($\alpha$TNF, 10 mg kg$^{-1}$) or vehicle at days 5 and 7, and with anti-PD-1 antibodies ($\alpha$PD-1, 10 mg kg$^{-1}$) or vehicle (PBS) at day 7. At day 10, CD8+ TILs (**c**) and the proportion of cell death in CD8+ TILs (**d**) were analysed by flow cytometry. Data from at least 5 tumors per group are represented as Tukey boxes (Mann–Whitney U test: *$p < 0.05$; **$p < 0.01$)

stabilization[23] and expression[15]. Accordingly, we observed a strong positive correlation between *TNFA* and *PDCD1LG1* expression in human melanoma specimens. Reduction of PD-L1 likely contributes to PD-1+ TIL accumulation in TNF-deficient mice. Infiltration of human melanoma tumors by CD8+ T cells, which co-express PD-1 and CTLA-4 at high levels and produce IFN-γ efficiently but not TNF, has been proposed as a good predictive marker of anti-PD-1 therapy[35]. The latter study suggested that TNF is unlikely required for, but rather impairs, the therapeutic response to anti-PD-1, a conclusion that agrees with our findings.

We also identified TNF as a factor required for anti-PD-1-induced TIM-3 expression on TILs, which critically impairs the therapeutic response to anti-PD-1 in mouse and human cancers[22,28,36,37]. Our study demonstrates that TNF potently induced TIM-3 expression on CD8+ TILs co-cultured with autologous melanoma cells from metastatic melanoma patients. We also found that the levels of the transcripts coding for TNF and TIM-3 positively correlate in human melanoma biopsies from the TCGA data bank as well as in tumors from anti-PD1-treated melanoma patients[23], further reinforcing the close molecular link between TNF and TIM-3 expression. Of note, TNF blockade or deficiency did not impair basal expression of TIM-3 on TILs indicating that other cytokines, such as IL-2, IL-7, IL-15, IL-21 or IL-27[31,38],

favor TIM-3 expression. Whereas the inhibition of TIM-3 expression by TILs upon anti-PD-1 therapy may contribute, at least in part, to the synergism of PD-1 and TNF blockade, additional mechanisms are likely involved. Indeed, as reported by others, TNF can trigger melanoma dedifferentiation as well as expression of the Ezh2 histone methyltransferase and the CD73 ectonucleotidase in melanoma cells, all of those events conferring resistance to immunotherapy[16,39,40]. Herein, we noticed that anti-PD-1 therapeutic effect was dramatically enhanced in TNFR1-deficient mice, indicating that TNFR1-dependent TNF signalling in the tumor microenvironment also critically contributes to anti-PD-1 therapy resistance. As a matter of fact, TNF-induced TIM-3 expression on activated T cells was impaired by TNFR1 deficiency.

In the context of a chronic viral infection, TNF induced CD4+ T cell dysfunction, also called T cell exhaustion, through PD-1 up-regulation[11]. In our study however, anti-PD-1 treatment induced TNF-dependent up-regulation of TIM-3, but not PD-1, on activated CD4+ and CD8+ TILs. To evaluate in more details the phenotype of the TIM-3+ TILs, we monitored IFN-γ and granzyme B expression in CD8+ TILs. TIM-3 expressing CD8+ TILs did not exhibit an exhaustion phenotype but rather potently produced IFN-γ and granzyme B under our experimental conditions. This indicates that, 3 days after a single anti-PD-1

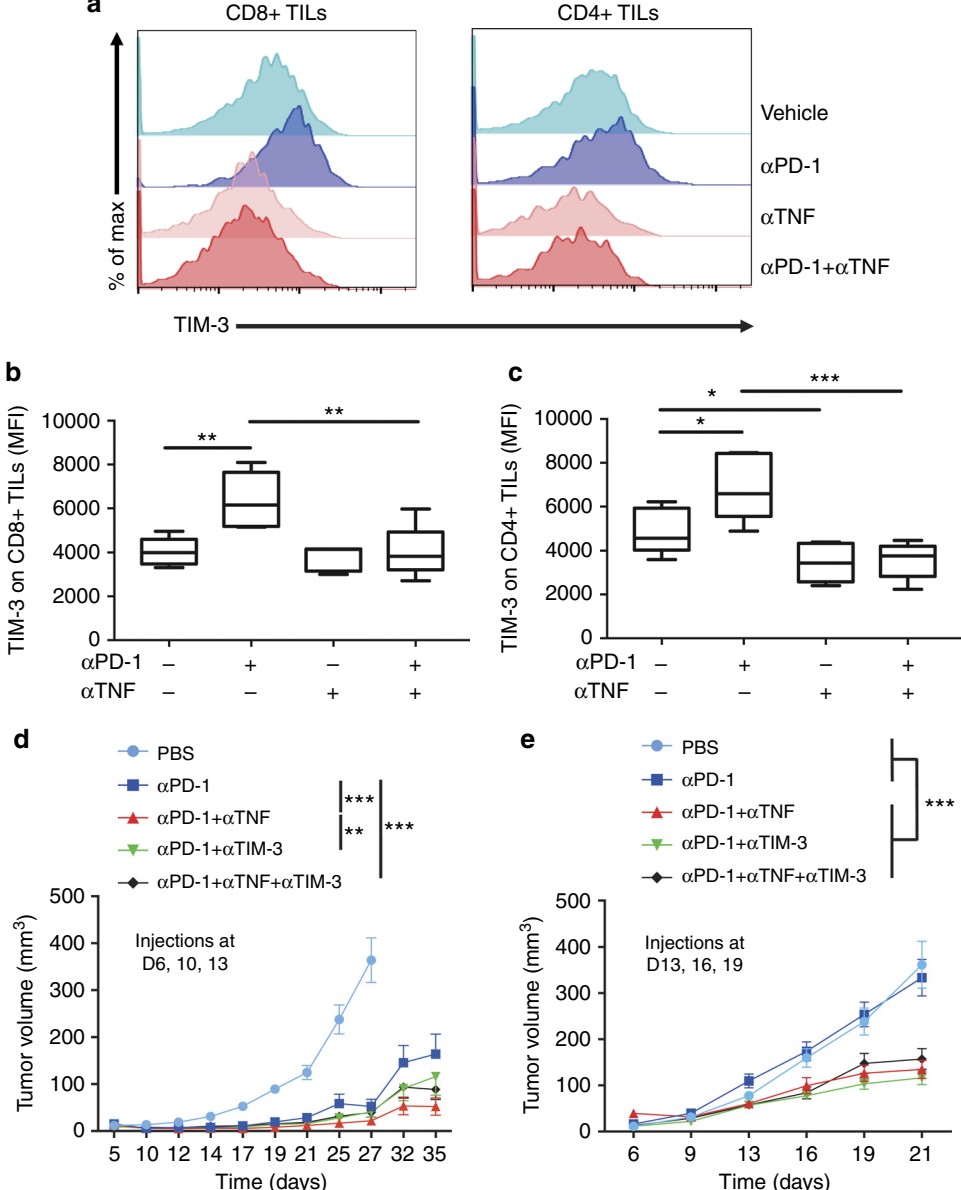

**Fig. 6** TNF blockade prevents TIM-3 up-regulation on TILs in response to anti-PD-1. WT mice were injected as described in the legend to Fig. 5. **a–c** TIM-3 expression level on CD8+ TILs and CD4+ TILs was determined by flow cytometry on tumors from WT mice at day 10 after B16K1 graft following injection of vehicle (PBS), anti-PD-1 (αPD-1), anti-TNF (αTNF) or a combination of both. Representative stainings (**a**) mean fluorescence intensity (MFI) of TIM-3 on CD8+ (**b**) and CD4+ TILs (**c**) measured in at least 5 tumors per group from one experiment are represented as Tukey boxes. **b**, **c**: Mann–Whitney U test: *$p < 0.05$; **$p < 0.01$; ***$p < 0.001$). **d** and **e** C57BL/6 WT mice were intradermally and bilaterally grafted with $3 \times 10^5$ B16K1 melanoma cells prior to injection with vehicle, anti-PD-1 alone or combined with anti-TNF and/or anti-TIM-3 (10 mg kg$^{-1}$ of each antibody) at days 6, 10, and 13 (d) (PBS ($n = 12$), αPD-1 ($n = 9$), αPD-1 + αTNF ($n = 14$), αPD-1 + αTIM-3 ($n = 13$), αPD-1 + αTNF + αTIM-3 ($n = 15$), data from two experiments) or at days 13, 16, and 19 (**e**) ($n = 10$ mice per group, data from one representative experiment out of two). Tumor volumes were determined with a calliper. Data are means ± s.e.m. (d) and e, two-way Anova: **$p < 0.01$, ***$p < 0.001$ at day 35 (**d**) and day 21 (**e**); $p = 0.08$ at day 35 when comparing anti-PD-1 alone and anti-PD-1 plus anti-TIM-3 (**d**)

injection, TIM-3 reflects the activation of TILs. It is likely, however, that TIM-3 upregulation triggers TIL exhaustion at later time points in WT mice, as reported by others[22,36]. The complete tumor regression in most of the TNF-deficient mice precluded analysis of TILs at later time points under our experimental conditions. Importantly, TIM-3+, but not TIM-3-,CD8+ TILs died upon anti-PD-1 therapy in a TNF-dependent manner, strongly supporting the notion that TNF triggered AICD in CD8 + TILs.

Whereas TNF deficiency modified neither CD8+ and CD4+ TIL proliferation nor chemokine expression in tumors, it reduced

cell death of CD8+ and CD4+ TILs. This was associated with an increased plasma membrane permeability, reflecting the permanent loss of plasma membrane barrier function and indicates that TILs indeed died, according to recent criteria for cell death[41]. Strikingly, anti-TNF reduced CD8+ TIL cell death upon anti-PD-1 therapy, further emphasizing the pivotal role of TNF in AICD of CD8+ T cells, in agreement with a previous study[13]. The TNF-induced up-regulation of TIM-3 and PD-L1 may also contribute to AICD of CD8+ TILs in TNF-proficient animals. Indeed, cell death specifically occurred in PD-1+ TIM-3+ CD8+ TILs upon anti-PD-1 therapy in WT animals.

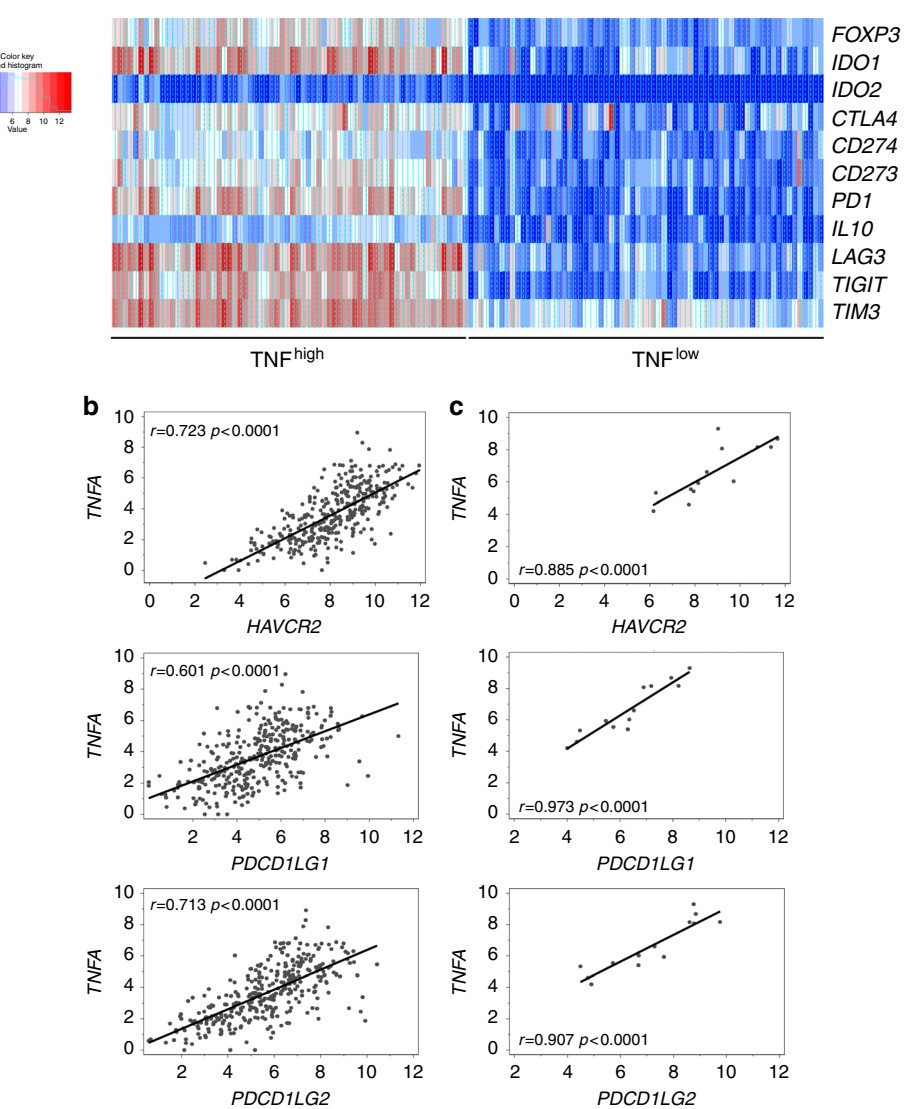

**Fig. 7** TNF expression is associated with an immune escape gene signature in human metastatic melanoma. **a** Heatmap for a selected list of genes encoding immune escape proteins in metastatic melanoma patients from the TCGA melanoma cohort ($n = 342$), exhibiting high (80th percentile) and low (20th percentile) TNF expression in melanoma samples. Genes were clustered using a Euclidean distant matrix and average linkage clustering. **b** and **c** Correlation analysis between the expression of *HAVCR2* (encoding TIM-3), *PDCD1LG1* (encoding PD-L1), *PDCD1LG2* (encoding PD-L2) and *TNFA* (encoding TNF) in melanoma samples from metastatic melanoma patients from the TCGA cohort ($n = 342$) (**b**) and from patients treated with anti-PD-1 (our analysis of data published in ref.[23]) ($n = 13$) (**c**)

TNF is considered to be an effector molecule of the cytotoxic T lymphocytes in anti-cancer immune response[5]. Here, we provide evidence that TNF is unlikely a key cytotoxic molecule in three murine cancer models upon anti-PD-1 therapy but rather compromises CD8+ TIL survival. Moreover, the loss of TNF did not affect the expression of granzyme B, another key effector molecule of CD8+ T-cell mediated cytotoxicity. Instead of being cytotoxic towards melanoma cell lines in vitro, TNF was shown to trigger melanoma dedifferentiation, contributing to inflammation-induced immunoediting processes and immune escape[16].

Anti-TNF antibodies, such as infliximab, are already used in the clinic to efficiently cure some irAEs associated with anti-PD-1 or anti-CTLA-4. However, these are neither concomitantly nor systematically administered with anti-PD-1 but only after discontinuation of immunotherapy in patients with severe immune-related colitis[24]. Our study shows that anti-PD-1 triggers TNF production, which, in turn, potently impairs CD8+ TIL response. Co-administering anti-PD-1 and anti-TNF may facilitate a sustained CD8+ T cell-dependent immune response by attenuating TNF-dependent AICD of CD8+ TILs and expression of TIM-3 and PD-L1. Taken together, our results provide the first proof-of-concept of combining anti-PD-1 and anti-TNF to fight melanoma and putatively other cancers. An ongoing phase 1b clinical trial (NCT03293784) will evaluate the safety and tolerance of the combination of immune checkpoint inhibitors and anti-TNF in metastatic melanoma patients.

## Methods

**Reagents and antibodies.** Anti-PD-1 (clone RMP1–14; 10 mg kg$^{-1}$), anti-TNF (clone XT3.11; 10 mg kg$^{-1}$), anti-TIM-3 (clone RMT3-23; 10 mg kg$^{-1}$) and isotype control antibodies (clone 2A3; 10 mg kg$^{-1}$) were purchased from BioXcell; anti-TNFR1 (clone 55R-170; 10 mg kg$^{-1}$) was from Ozyme. Additional antibodies used in this study were anti-mouse CD45 (BD Biosciences, BUV395, clone 30-F11; 1/200), anti-mouse Thy1 (Biolegend, APC-Cy7, clone 30-H12; 1/400), anti-mouse

CD8 (BD Biosciences, BV605, clone 53-6.7; 1/200), anti-mouse CD4 (BD Biosciences, BUV496, clone GK1.5; 1/200), anti-mouse TIM-3 (eBioscience, PE-Cy7, clone RMT3-23; 1/200), anti-mouse LAG3 (Biolegend, PE, clone C9B7W; 1/100), anti-mouse PD-1 (eBioscience, FITC, clone J43; 1/200), anti-mouse CTLA-4 (eBioscience, APC, clone UC10-4B9; 1/100), anti-mouse TIGIT (Biolegend, Pe-Cy7, clone 1G9; 1/100), anti-mouse CD11c (eBioscience, PE-Cy7, clone N418; 1/100), anti-mouse CD3 (Biolegend, FITC, clone 145-2C11; 1/200), anti-mouse PD-L1 (BD Biosciences, PE, clone MIH5; 1/100), anti-mouse PD-L2 (BD Biosciences, APC, clone TY25; 1/100), anti-mouse Ki67 (BD Biosciences, FITC, clone B56; 1/10), anti-human TIM3 (BD Biosciences, BV421, clone 7D3; 1/50) and anti-human CD8 (BD Biosciences, BV605, clone SK1; 1/50). Cell suspensions were incubated with CD16/CD32 blocking antibodies (eBioscience; 1/200) prior to incubation with fluorochrome-conjugated antibodies. For analysis of live and dead cells, the LIVE/DEAD fixable Aqua reagent (Invitrogen) was used.

**Cell lines.** B16K1 is a genetically modified cell line generated from B16F10 cells, and that stably expresses the MHC-I molecule H-2Kb, known to stimulate CD8+ T cell-dependent immune responses[42,43]. Cells were cultured in DMEM medium containing 10% heat-inactivated foetal calf serum (FCS) and were authenticated in February 2012 by the Leibniz-Institut DSMZ GmbH. To guarantee cell line authenticity, the B16K1 cell line was used for a limited number of passages and tested for the expression of melanocyte-lineage proteins such as tyrosinase-related protein 2 (TRP2). Medium was changed every 2–3 days. Lewis lung carcinoma (LLC) and 4T1 breast cancer cells were from ATCC; they were not further authenticated but cultured for a limited number of passages in DMEM medium containing 10% FCS. Cell lines were tested for the absence of mycoplasma contamination by PCR.

**Animal models.** TNF-deficient (#5540) and TNFR1-deficient (#2818) C57BL/6 mice were purchased from Jackson laboratories. CD8-deficient C57BL/6 mice were a gift from Prof. J. van Meerwijk (INSERM U1043, Toulouse, France). C57BL/6 and Balb/c mice were from Janvier and Envigo, respectively. Mice had unrestricted access to food and water and were kept on a 12-h light/dark cycle under specific pathogen-free conditions at the CRCT animal facility (US006 CREFRE—Inserm/UPS), which is accredited by the French Ministry of Agriculture to perform experiments on live mice (accreditation number A-31 55508). All experimental protocols were approved by the local ethic committee (Midi-Pyrénées, France) and are in compliance with the French and European regulations on care and protection of laboratory animals. For each experimental condition, at least 5 female 6–12-week-old mice were used.

**In vivo tumorigenesis.** $3 \times 10^5$ or $1 \times 10^6$ B16K1 and $4 \times 10^5$ LLC cells were intradermally and bilaterally injected in WT, TNF-deficient and TNFR1-deficient mice. Anti-PD-1 (10 mg kg$^{-1}$) or isotype control (10 mg kg$^{-1}$) was next intraperitoneally injected at days 6, 10 and 13 after B16K1 graft. Anti-TNF[44] (10 mg kg$^{-1}$) was inoculated at days 6, 10, 13 and 16 and anti-TNFR1[44] (10 mg kg$^{-1}$) at days 6 and 10. Alternatively, mice were injected with relevant isotype control antibodies (10 mg kg$^{-1}$). Tumor volumes were calculated using a caliper at the indicated days with the formula: tumor volume = $0.52 \times$ length $\times$ width$^2$.

**Overall survival.** The overall survival was estimated taking into account the time of natural death of animals or, alternatively, the time of sacrifice when tumors reached 10% of body weight (i.e., total tumor volume of 2000 mm$^3$), according to national and international policies.

**In vivo tumorigenesis upon anti-PD-1, anti-TNF and anti-TIM-3 injections.** Experiments were performed to evaluate the contribution of TIM-3 in mediating the synergism observed upon anti-PD-1 and anti-TNF injections in two experimental settings:

Protocol 1: C57BL/6 WT mice were bilaterally grafted with $3 \times 10^5$ B16K1 melanoma cells prior to injection with vehicle, anti-PD-1 alone or combined with anti-TNF and/or anti-TIM-3 at days 6, 10, and 13 (10 mg kg$^{-1}$ for each antibody). Protocol 2: Similar experiment was performed as for protocol 1 with antibody injections carried out at days 13, 16, and 19.

**4T1 breast cancer cell growth in mice.** $1 \times 10^5$ 4T1 cells were orthotopically injected into the mammary fat pad of Balb/c mice. Mice were next injected with isotype control, anti-PD-1, anti-TNF or the combination of anti-PD-1 and anti-TNF (10 mg kg$^{-1}$ of each antibody) at days 6, 9 and 13. Tumor volume was calculated using a caliper at the indicated days with the formula: Tumor volume = $0.52 \times$ length $\times$ width$^2$.

**Analysis of the lymphocyte content of tumors.** $1.10^6$ B16K1 cells were intradermally injected in WT and TNF-deficient mice. At day 7, mice were injected with anti-PD-1. At day 10, mice were sacrificed and tumors were collected, weighed and digested with the Tumor Dissociation Kit, mouse (Miltenyi). Cells were stained with the indicated antibodies and LIVE/DEAD reagent before flow cytometry analysis (BD LSRFortessa X-20).

**Analysis of tumor transcripts.** $10^6$ B16K1 cells were intra-dermally injected in WT, TNF-deficient and CD8-deficient mice. Anti-PD-1 (10 mg kg$^{-1}$) or vehicle (PBS) was next intraperitoneally injected at day 7. At day 10, mice were sacrificed and tumors were collected and dissociated using a Precellys Evolution tissue homogenizer (Bertin technologies) at 8000 rpm for 2 cycles of 30 s in vials containing ceramic balls. RNA was purified using the RNeasy Midi Kit (Qiagen). cDNA was prepared with SuperScript II reverse transcriptase (Thermofischer) using 1 µg of total RNA from each sample. qPCR was performed using SYBR Green Master Mix (Takara) and primers for transcripts encoding murine β-actin, HPRT, TNFα, IFN-γ and chemokines (Qiagen).

**Analysis of TIM-3 induction on purified lymphocytes.** Murine naive CD8+ T cells were purified from the spleen of naive WT or TNFR1-deficient mice using a mouse naive CD8+ T cell purification kit (Miltenyi Biotec). Human mononuclear cells were obtained from the peripheral blood of healthy donors (Etablissement Français du Sang, Centre Hospitalier Universitaire de Toulouse, France) after informed, signed consent. Human naive CD8+ T cells were isolated using a human naive CD8+T cell purification kit (Miltenyi Biotec). Murine and human T cells were activated with anti-CD3 and anti-CD28-coated beads (Life Technologies) and anti-CD3, anti-CD28 and anti-CD2 coated beads (Miltenyi), respectively, in the presence of IL-2 (Invitrogen; 200 U mL$^{-1}$). Activated CD8+ T cells were incubated for 48 h in the presence of either recombinant murine or human TNF (Peprotec) and TIM-3 expression was then evaluated by flow cytometry.

**Analysis of TIM-3 expression in autologous human co-culture experiments.** Protocol was approved by "CPP du sud-ouest et outre-mer IV" (Limoges). Informed, signed consents from metastatic melanoma patient 1 and patient 2 were obtained. Patient 1 was affected with metastatic acrolentiginous melanoma and primary tumor sample was collected before therapy induction. We obtained from patient 2, who was diagnosed with metastatic cutaneous melanoma and treated with Nivolumab and Ipilimumab, a biopsy of one abdominal metastasis. Tumor samples from the two metastatic melanoma patients were subjected to mechanical dilacerations followed by filtration through a 40 µm pore strainer. The stroma vascular fraction was then plated overnight to allow for the adhesion and expansion of tumor cells in DMEM 10% FCS. The next day, non-adherent cells were subjected to Ficoll (Ficoll-Paque$^{TM}$ Plus, GE Healthcare) separation. Cells at the interphase ring were collected and plated for expansion in DMEM (Lonza) containing 10% FCS, 2 mM L-glutamine (Sigma), 50 µM 2-β-mercaptoethanol (Sigma), 10 mM Hepes (Gibco), 10 U ml$^{-1}$ Penicillin (Sigma), 0.1 mg ml$^{-1}$ Streptomycin (Sigma) and 6,000 U ml$^{-1}$ human recombinant IL-2 (Proleukin) for 10 days before being frozen down.

When primary melanoma tumor cells from each patient were sufficiently expanded, lymphocytes were re-stimulated in the presence, or not (IL-2 alone), of autologous melanoma cells at a ratio of 20:1 for 48 h in the presence of 200 U ml$^{-1}$ IL-2, with or without recombinant human TNF. TIM-3 expression on CD8+ TILs was evaluated by flow cytometry. For co-culture experiments with TILs from patient 2, the number of TILs extracted allowed us to test the impact of various concentrations of TNF on TIM-3 expression.

**Analysis of TNF, TIM-3, PD-L1/2 expression in human melanoma.** TNF expression was analysed using the TCGA melanoma cohort. TCGA genomic and clinical data were downloaded from the UCSC cancer genome browser project (https://genomecancer.ucsc.edu). The analysis population consisted of 342 patients with distant metastasis for whom RNAseq and clinical data overlap. No formal sample size calculation was performed concerning TCGA analysis. Gene expression was measured experimentally using the Illumina HiSeq 2000 RNA Sequencing platform and log2(x + 1) transformed. Alternatively, TNF expression was analysed in melanoma biopsies from patients treated with anti-PD-1 (data published in ref. 23). The strength of the relationship between TNF, TIM-3, PD-L1 and PD-L2 encoding genes was assessed using Spearman's rank correlation coefficient.

**Statistics and reproducibility.** Each study was designed to use the minimum number of mice or samples required to obtain informative results and sufficient material for subsequent studies. No specific statistical test was used to predetermine the sample size. For animal experimentation, we used at least 5 mice and experiments were typically performed twice, unless otherwise stated in the figure legends. Statistical significance of difference between groups was evaluated by using the Graph-Pad Prism 7 software. Briefly, we tested whether the values come from a Gaussian distribution using a D'agostino-Pearson omnibus normality test (for sample sizes over 7) or the Shapiro Wilk (for samples sizes under 7). When passing the normality test, a two-sided Student's t-test was used. Otherwise, a Mann–Whitney U test was used. For statistical significance of animal survival, the log-rank test was used. Differences were considered to be statistically significant

when $p < 0.05$ (*$p < 0.05$; **$p < 0.01$; ***$p < 0.001$). In vivo and in vitro experiments were monitored in a non-blinded fashion, no method of randomization was used and usually no sample was excluded for the analysis. For the analysis of TILs, tumors in which the number of cells was too small were excluded from the analysis. Global growth curves and, when indicated, flow cytometry data were analysed using a Two-way Anova test with Tukey multiple comparison test.

**Data availability**. The data that support the findings of this study are available from the corresponding author upon reasonable request.

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

## Acknowledgements

The results shown here are in whole or part based upon data generated by the TCGA Research Network: http://cancergenome.nih.gov/. We are grateful to the flow cytometry, microscopy and animal facilities of the I2MC (INSERM U1048, Toulouse, France) and CRCT (INSERM U1037, Toulouse, France) for their technical assistance. We thank Prof. J. van Meerwijk (INSERM U1043, Toulouse, France) for the kind gift of CD8-deficient mice. We thank Drs. J.J. Fournié, J. Riond, L. Martinet, M. Ayyoub (INSERM U1037, Toulouse, France) and A.F. Tilkin-Mariamé (INSERM U1220, Toulouse, France) for fruitful discussion. This work was supported by Ligue nationale contre le cancer, Ligue régionale contre le cancer, Cancéropôle Grand Sud-Ouest, INSERM Transfert, Institut National du Cancer, ROTARY Toulouse clubs, Fondation Toulouse Cancer Santé, INSERM and Paul Sabatier University (Toulouse III). Prestige co-financing grant award (REA grant agreement n. PCOFUND-GA-2013-609102).

## Author contributions

F.B., A.M., C.C., E.M., and C.I. designed and performed preclinical experiments; P.R. wrote the protocol for experiments on human melanoma samples, obtained informed consent from patients and supervised tissue collection processes; A.M. performed autologous co-culture experiments; J.G. and T.F. performed biostatistical analyses; F.B., A.M., C.C., E.M., C.I., N.A.A., T.L. and N.M. analysed the data and edited the paper; C.C. and B.S. designed and supervised the study and wrote the paper.

## Additional information

**Competing interests:** Nicolas Meyer has worked as an investigator and/or consultant and/or speaker for: BMS, MSD, Amgen, Roche, GSK, Novartis, Pierre Fabre. The rest of the authors declare no competing financial interests.

