## [Peer Review File · Nature Communications]

Reviewers' comments:

Reviewer #2 (Remarks to the Author):

Overall, the present manuscript presents data in mouse tumor models supporting the additive effects of PD-1 blockade and TNF blockade in three mouse tumor models. This study follows on the previous study by the same authors showing the CD8+ T-cell-mediated antitumor effects of TNF blockade in mouse tumor models.

Using one mouse tumor melanoma model with MHC-transfected melanoma cells, the investigators shows that in the absence of TNF, CD8+ T cell infiltrates augment, with decreased T cell death. The novelty in this manuscript is to show the additive effects of PD-1 blockade and TNF blockade/TNF deletion and the role of TNF in upregulating Tim-3 expression by T cells. PD-1 blockade did not further augment the number of CD8+ TILs in TNF^{-/-} mice but promote the increase of IFN- γ and granzyme production.

MAJOR COMMENTS.

1. Overall, the observed effects are made in two immunogenic mouse tumor models. The melanoma cell line used in this study has been transfected with MHC class I. The Lewis lung cancer model is also a very immunogenic tumor model and although MHC class I may be low, the possibility of antigen presentation by non-polymorphic MHC class I has been reported. It is therefore not surprising that PD-1 blockade works well in inflamed tumors. The major question would be to determine whether TNF blockade may convert a non-inflamed tumor into an inflamed tumor to increase the clinical efficacy of PD-1 blockade. This is not addressed in the present manuscript. Notably, the effects of dual TNF/PD-1 blockade in breast cancer tumor model are modest.
2. The observation that TNF deficiency impedes Tim-3 upregulation upon PD-1 blockade in CD8+ TILs is interesting and one would like to know whether such effects are observed upon TNF blockade in CD8+ TILs. Also, the effects of TNF would need to be evaluated in the absence of TCR activation to dissect the effects of TNF and TCR stimulation, respectively. Most, importantly, the investigators do not show convincing evidence that effects of the dual PD-1/TNF blockade are mediated by Tim-3 expression levels.
3. Notably, TNF has been shown to contribute to T cell exhaustion of antigen-specific CD8+ T cells in chronic viral infections (Beyer et al., *Nature Immunol*, 2016). One would like to see an in-depth evaluation of the functional capacities of tumor antigen-specific CD8+ TILs.
4. The immunological effects are evaluated only after three days following PD-1 injection. One wonders what fraction of tumor-infiltrating CD8+ T cells is truly tumor-reactive.
5. The evaluation of T cell death in TILs compared PD-1 blockade in WT vs. TNF^{-/-} mice and one like to also see the data in TNF^{-/-} mice.
6. The evaluation of Tim-3 in TILs should present both frequencies and MFI. Also as Tim-3 is a marker of T cell activation, it is not surprising to see Tim-3 upregulation upon PD-1 blockade. This has already been reported in vitro (Fourcade et al., *J Exp Med*, 2010). One would like to see the evaluation of PD-1+Tim3+ CD8+ T cell function in each experimental condition to support

potential T cell dysfunction vs. T cell activation.

Reviewer #3 (Remarks to the Author):

The authors have done a series of experiments addressing all of my commentaries. In my opinion they have also addressed some of the most relevant comments from other reviewers. In doing this they have significantly improved the quality and conclusions of this manuscript. They've added data to support the connection between TNF α and TIM3 expression and expanded on the mechanism by which TNF blockade could be impacting on anti-tumour immunity, and T cell activation and survival within the tumour. A Key experiment is shown in fig A to reviewers and 8D in new manuscript which shows that anti TIM3 does not add to anti PD1+anti TNF thus supporting the notion that TNF blockade is acting at least in part by affecting TIM3 expression. This experiment has only be done once and furthermore it is not clear to me why the protocol for intervention with antibodies (d13,17,20) is different from the other tumour rejection experiments (6,9,13,16) in figure 7. If this is such a relevant point for the conclusion then this data must be further validated.

We thank the Editor and Reviewers for their constructive evaluation that greatly helped us to improve our manuscript.

As recommended by the Reviewers, the molecular and cellular mechanisms that underlie our novel observations have been further investigated. The following letter includes a point-by-point response to their comments.

Reviewers' comments:

Reviewer #2 (Remarks to the Author):

Overall, the present manuscript presents data in mouse tumor models supporting the additive effects of PD-1 blockade and TNF blockade in three mouse tumor models. This study follows on the previous study by the same authors showing the CD8+ T-cell-mediated antitumor effects of TNF blockade in mouse tumor models.

Using one mouse tumor melanoma model with MHC-transfected melanoma cells, the investigators shows that in the absence of TNF, CD8+ T cell infiltrates augment, with decreased T cell death. The novelty in this manuscript is to show the additive effects of PD-1 blockade and TNF blockade/TNF deletion and the role of TNF in upregulating Tim-3 expression by T cells. PD-1 blockade did not further augment the number of CD8+ TILs in TNF^{-/-} mice but promote the increase of IFN- γ and granzyme production.

An important issue to consider is that our data are not simply additive effects of PD-1 and TNF blockade. They rather indicate synergism between both immunotherapies. For instance, as depicted in new Figure 5a, TNF blockade alone did not trigger tumor regression (0/24 tumor regression); anti-PD-1 alone induced only 30% tumor regression (7/24 tumor regression); anti-TNF + anti-PD-1 induced 75% tumor regression (18/24 tumor regression).

Very similar data were obtained in TNF-deficient mice upon anti-PD-1 therapy as depicted in Fig. 1.

Novel experiments now revealed the critical role of TNF in anti-PD-1-triggered AICD (Activation-induced cell death) in CD8+ T cells, as documented both in TNF-deficient mice (Fig. 3c, right panel; Supplementary Fig. 11c) as well as in WT mice treated with anti-TNF (Fig. 6c).

To the best of our knowledge, this finding is new and original in the context of PD-1 blockade therapy. Thus, we identify herein TNF-induced AICD, which is triggered upon anti-PD-1, as a potent immune escape mechanism, limiting CD8+ TIL accumulation and contributing to anti-PD-1 resistance in inflamed tumors.

MAJOR COMMENTS.

1. Overall, the observed effects are made in two immunogenic mouse tumor models. The melanoma cell line used in this study has been transfected with

MHC class I. The Lewis lung cancer model is also a very immunogenic tumor model and although MHC class I may be low, the possibility of antigen presentation by non-polymorphic MHC class I has been reported. It is therefore not surprising that PD-1 blockade works well in inflamed tumors. The major question would be to determine whether TNF blockade may convert a non-inflamed tumor into an inflamed tumor to increase the clinical efficacy of PD-1 blockade. This is not addressed in the present manuscript. Notably, the effects of dual TNF/PD-1 blockade in breast cancer tumor model are modest.

We agree with the reviewer that one of the limitations to the successful use of ICI to treat cancer patients is the requirement for tumors to be pre-infiltrated with immune cells (i.e., inflamed tumors) and more specifically TILs. However, a high proportion of patients affected by cancers that are described as “immunogenic” still poorly respond to these therapies. As opposed to the “primary resistance”, these resistance mechanisms are defined as “adaptive immune resistance” and “acquired resistance” (PMID: 28187290). One documented cause for this phenomenon is the increased expression of secondary immune checkpoint molecules such as TIM-3 that dampen the anti-tumor immune response (PMID: 26883990). This is now discussed page 12.

In the present manuscript, we demonstrate for the first time to our knowledge that anti-TNF antibodies overcome the resistance of cancers to anti-PD-1 therapy in the following three distinct cancer models.

The mouse melanoma B16K1 model (see Figs. 5a and b):

B16K1 is an immunogenic melanoma model due to the expression of MHC class I at high levels. However, one should note that only 30% of B16K1 tumors totally regressed upon anti-PD-1 therapy under our experimental conditions. This is very close to the rate of patients affected with metastatic melanoma who respond well to anti-PD-1, indicating that our model has clinical relevance (see refs #19 and 20). Whereas anti-TNF antibodies alone substantially reduced B16K1 tumor growth, they could not lead to total tumor regression. Moreover, combination of anti-TNF and anti-PD-1 dramatically increased the frequency of total regression (i.e., 75%). Thus, the effects of anti-TNF and anti-PD-1 in these cancer models are not additive but synergistic, preventing the tumor relapses that are observed in about 15% of advanced melanoma patients during the 2-year post-induction.

In addition, mice, which totally rejected B16K1 tumors, did not develop tumor growth upon a second B16K1 cell injection. Thus, TNF is not required for the establishment of a long-lasting immune response against melanoma.

The Lewis lung carcinoma (LLC) model (Supplementary Fig. 3):

Although the Lewis lung cancer model may express non-polymorphic MHC class I molecules, this model failed to respond to anti-PD-1 therapy in wild-type mice under our experimental conditions. As stated in the manuscript, whereas TNF or TNF-R1 deficiency did not impair LLC growth, they both overcame the resistance to anti-PD-1.

The 4T1 breast cancer model (Supplementary Fig. 13):

Our observation was next extended to another cancer model, the 4T1 breast cancer cell line, which was orthotopically implanted in Balb/c mice (i.e., a genetic background different from that of the two other models). This model failed to respond to anti-PD-1 or anti-TNF alone. However, their combination triggered significant efficacy.

Collectively, our observations indicate synergistic effects of TNF blockade and anti-PD-1 in three different “inflamed”, yet totally or partially resistant to anti-PD-1 therapy, experimental cancer models.

Whether these synergistic effects can occur in advanced melanoma patients will be shortly evaluated in our institute in a clinical trial (confidential information).

2. The observation that TNF deficiency impedes Tim-3 upregulation upon PD-1 blockade in CD8+ TILs is interesting and one would like to know whether such effects are observed upon TNF blockade in CD8+ TILs. Also, the effects of TNF would need to be evaluated in the absence of TCR activation to dissect the effects of TNF and TCR stimulation, respectively. Most, importantly, the investigators do not show convincing evidence that effects of the dual PD-1/TNF blockade are mediated by Tim-3 expression levels.

We agree with the reviewer it is essential to evaluate the impact of TNF deficiency on tumor growth in a way that would be more relevant to the clinic using anti-TNF antibodies in addition to experiments in the TNF $-/-$ mice. Experiments using anti-TNF antibodies were shown in Figures 7 and 8 of the previous version of the manuscript. We are sorry this did not stand out in the previous version; we amended the text pages 10-11 to make it clearer.

To show that TNF can promote TIM-3 expression in the absence of TCR activation, we purified naïve CD8+ T cells from the spleen and lymph nodes of C57Bl/6 mice. To be consistent with the experiment depicted in the manuscript, the cells were cultured for 5 days with IL-2 alone. At day 5, cells were treated with increasing concentrations of murine TNF for 48h. Although the effect is less pronounced in the absence of TCR engagement, the Figure A below shows that TNF is able to promote TIM-3 upregulation on naïve CD8+ T cells. This is now indicated as data not shown in the “results” section page 9.

Figure A (not to be included in the manuscript). TNF increases TIM-3 up-regulation on murine CD8+ T cells in the absence of TCR engagement. Naïve and unstimulated CD8+ T cells were incubated with the indicated TNF concentrations for 48 hours. TIM-3 expression was analysed by flow cytometry.

We also performed additional experiments with TILs isolated from two human metastatic melanoma patients (see Figures 4b and c and supplementary Fig 12d). TNF triggered TIM-3 up-regulation on TILs and this effect was even more pronounced upon co-culture in the presence of autologous melanoma cells.

Finally, as requested by Reviewer #3, we performed an additional experiment by co-administering anti-PD-1 with anti-TIM-3 or anti-TNF, with the first injection at day 6 following B16K1 graft. Whereas the combination of anti-TIM-3 and anti-PD-1 significantly impaired tumor growth as compared to vehicle, the effect was significantly less potent than that of anti-PD-1 and anti-TNF therapy (Fig. 6d). Moreover, the addition of anti-TIM-3 did not potentiate the efficacy of combining anti-PD-1 and anti-TNF (Figs 6d and e). Thus, the data indicate that the therapeutic benefit of combining anti-TNF and anti-PD-1 is not only restricted to its impact on the inhibition of the TIM-3-dependent pathway. As mentioned in the present version of the manuscript, we provide evidence that anti-PD-1 triggered TNF-dependent PD-L1 expression and AICD of TILs. These mechanisms also likely contribute to the therapeutic effects of combining anti-PD-1 and anti-TNF.

3. Notably, TNF has been shown to contribute to T cell exhaustion of antigen-specific CD8+ T cells in chronic viral infections (Beyer et al., Nature Immunol, 2016). One would like to see an in-depth evaluation of the functional capacities of tumor antigen-specific CD8+ TILs.

In the context of a chronic viral infection, TNF-induced CD4+ T cell dysfunction through PD-1 up-regulation has indeed been shown (PMID: 26950238). This reference is now included in the discussion section of the manuscript page 13. In our study however, we show that anti-PD-1 treatment induces the up-regulation of another immune checkpoint, TIM-3, on activated CD8+ TILs and demonstrated the involvement of TNF in this process in mice.

To evaluate in more details the phenotype of the TIM-3+ TILs in our settings, we performed an additional experiment and monitored IFN- γ and granzyme B expression in CD8+ TILs in the various conditions (WT and TNF-deficient mice injected with vehicle or anti-PD-1). Data depicted in Supplementary Fig. 7 indicate that TILs are unlikely exhausted 3 days after one single anti-PD-1 injection. Indeed, the enhanced TIM-3 expression on TILs following anti-PD-1 treatment did not decrease the percentage of the IFN- γ or granzyme B producing CD8+ TILs.

Hence, we agree with the reviewer that under our experimental conditions TIM-3 can be considered as an activation marker. However, the expression of both PD1 and TIM-3 on TILs is well described as a characteristic feature of T cells that will undergo exhaustion (PMID: 20819927), including in the context of tumor development. This particular point is now developed in the discussion part page 13.

We performed additional experiments to monitor the functional capacities of tumor antigen-specific CD8+ TILs by using the dextramer technology. Unfortunately, as depicted below, we underwent technical difficulties and were dissatisfied with the staining we obtained with our new batch of dextramers. Indeed, the staining of TRP-2-specific CD8+ TILs appears to lack specificity, which precluded further analyses (Figure B). In our previous studies, the percentage of TRP-2-specific CD8+ TILs was globally 10 times lower than the one measured in this experiment. Although immunodominant, TRP-2 antigen is not overexpressed in B16K1 cells. Thus, it is unlikely the percentages depicted in Figure B truly reflect the amount of TRP-2-specific CD8+ TILs.

Figure B (not to be included in the manuscript): Analysis of TRP-2-specific CD8+ TILs in WT and TNF-deficient mice. C57BL/6 wild-type (WT) and TNF-deficient (TNF KO) mice were intradermally grafted with 1×10^6 B16K1 melanoma cells prior to intraperitoneal injection of anti-PD-1 antibodies (α PD-1, 10 mg/Kg) or vehicle (PBS) at day 7. At day 10, mice were sacrificed and TIL content was analysed by flow cytometry. The proportion of cells specific for TRP-2 among CD8+ TILs was determined. Data are means \pm sem of 5-6 tumors per group.

4. The immunological effects are evaluated only after three days following PD-1 injection. One wonders what fraction of tumor-infiltrating CD8+ T cells is truly tumor-reactive.

Please refer to our answer to item# 3.

As indicated above, the experiments performed to monitor TRP-2-specific CD8+ TILs could not be included in the manuscript. However, we indirectly addressed this point by evaluating the proportion of PD-1+ cells among CD8+ TILs. We consider PD-1+ CD8+ TILs represent the pool of T cells that reacted to antigens in the tumors (although only a fraction of PD-1+ CD8+ TILs may truly react to tumor-specific antigens). This proportion averaged 60% in the different groups under our experimental conditions, irrespectively of anti-PD-1 treatment and TNF status.

5. The evaluation of T cell death in TILs compared PD-1 blockade in WT vs. TNF^{-/-} mice and one like to also see the data in TNF^{-/-} mice.

Additional experiments have been performed to address this issue. As depicted in new Figure 3c (right panel) and 3d as well as new Supplementary Figure 11c, cell death was evaluated in WT vs TNF-deficient mice in both the control (i.e., PBS) and treatment (anti-PD-1) conditions.

The data show that basal CD8+ TIL cell death remains unchanged by TNF deficiency in control conditions and anti-PD-1-triggered CD8+ TIL AICD occurred in wild-type mice only and this was totally abrogated by TNF deficiency (Supplementary Fig. 11c).

6. The evaluation of Tim-3 in TILs should present both frequencies and MFI. Also as Tim-3 is a marker of T cell activation, it is not surprising to see Tim-3 upregulation upon PD-1 blockade. This has already been reported in vitro (Fourcade et al., J Exp Med, 2010). One would like to see the evaluation of PD-1+Tim3+ CD8+ T cell function in each experimental condition to support potential T cell dysfunction vs. T cell activation.

As mentioned above (see response to comment 5), additional experiments have been performed to evaluate the function and viability of PD-1+ TIM-3+ CD8+ TILs. As depicted in new Figs 3c and d, PD-1+ TIM-3+ CD8+ TILs were the only cells significantly affected by cell death following anti-PD-1 treatment; an effect abrogated by TNF deficiency. Considering TIM-3 and PD-1 as activation markers, cell death induced by anti-PD-1 was likely TNF-dependent AICD as stated in the manuscript.

To assess their activation status we also stimulated TILs with PMA/ionomycin in the presence of brefeldin A to simultaneously monitor IFN- γ and granzyme B production in CD8+ TILs expressing or not PD-1 and TIM-3. Under these experimental conditions, we lost the PD-1 staining on CD8+ TILs. However, the proportion of CD8+ TILs expressing IFN- γ and granzyme B was higher in TIM-3+ cells in each experimental condition (Supplementary Figs 11a and b). Strikingly, anti-PD-1-triggered CD8+ TIL cell death occurred in TIM-3+ cells only and this was totally abrogated by TNF deficiency (Supplementary Fig. 11c).

Reviewer #3 (Remarks to the Author):

The authors have done a series of experiments addressing all of my commentaries. In my opinion they have also addressed some of the most relevant comments from other reviewers. In doing this they have significantly improved the quality and conclusions of this manuscript. They've added data to support the connection between TNF α and TIM3 expression and expanded on the mechanism by which TNF blockade could be impacting on anti-tumour immunity, and T cell activation and survival within the tumour. A Key experiment is shown in fig A to reviewers and 8D in new manuscript which shows that anti TIM3 does not add to anti PD1+anti TNF thus supporting the notion that TNF blockade is acting at least in part by affecting TIM3 expression. This experiment has only be done once and furthermore it is not clear to me why the protocol for intervention with antibodies (d13,17,20) is different from the other tumour rejection experiments (6,9,13,16) in figure 7. If this is such a relevant point for the conclusion then this data must be further validated.

As requested, we performed an additional experiment by co-administering anti-PD-1 with anti-TIM-3 or anti-TNF, with the first injection at day 6 following B16K1 graft. Whereas the combination of anti-TIM-3 and anti-PD-1 significantly impaired tumor growth as compared to vehicle, the effect was significantly less potent than that of anti-PD-1 and anti-TNF therapy (Fig. 6d and Supplementary Fig. 14). Moreover, the addition of anti-TIM-3 did not potentiate the efficacy of combining anti-PD-1 and anti-TNF (Figs 6d and e). Thus, the data indicate that the therapeutic benefit of combining anti-TNF and anti-PD-1 is not only restricted to its impact on the inhibition of the TIM-3-dependent pathway. As mentioned in the present version of the manuscript, we provide evidence that anti-PD-1 triggered TNF-dependent PD-L1 expression and AICD of TILs. These mechanisms also likely contribute to the therapeutic effects of combining anti-PD-1 and anti-TNF.

Reviewers' comments:

Reviewer #2 (Remarks to the Author):

Overall, the synergy between PD-1 blockade and TNF depletion in the three mouse tumor models is not well determined. The authors show some evidence that TNF promotes some the upregulation of Tim-3 by T cells, some upregulation of PD-L1 by T cells and APCs and that in the absence of TNF, CD8+ TILs exhibit less cell death. However, it remains unclear how and why any of these observations can explain the antitumor effects observed upon therapy with PD-1 blockade and TNF depletion.

MAJOR COMMENTS.

1. The investigators show some data demonstrating the role of TNF in promoting a modest upregulation of Tim-3 by T cells in mouse and in TILs. In addition, using anti-PD1, anti-Tim-3, and anti-TNF blocking antibodies, the antitumor effects of dual PD1/TNF blockade appear superior. In these experiments, the role of Tim-3 in mediating some of the effects of TNF blockade is not demonstrated.

2. The experiments to evaluate TIL functions are made at the RNA level and upon stimulation with PMA/ionomycin. The investigators observed a slight decrease of IFN production by CD8+ TILs in the absence of TNF. They also report granzyme B expression by CD8+ TILs is in TNF^{-/-} mice, independently of PD-1 blockade. These observations are of uncertain significance and do not explain the strong synergistic effects of PD-1 blockade and TNF depletion. Also, one wonders about the CD8+ T cell-mediated tumor reactivity, which is not tested in these experiments.

3. The authors provide new data showing less T cell death in CD8+ TILs isolated from TNF^{-/-} mice treated with anti-PD-1 antibodies. The observation that PD-1 blockade would induce CD8+ TIL death goes against all the evidence that PD-1 regulates T cell survival and that PD-1 blockade increases T cell survival and function in the TME. They also show data suggesting that anti-PD-1 therapy triggered TNF-dependent AICD of PD-1+ Tim-3+ T cells. This effect is very modest as % cell death within PD-1+Tim3+ is at next 7-8 % vs. 3-5% in Tim-3-PD-1⁻. It is very unlikely that such small increase in % dead cells is responsible for the clinical effects of dual TNF/PD-1 blockade.

Overall, the mechanisms of action of the dual TNF/PD-1 blockade remain uncertain. A suggested by Beyer et al., TNF may play a critical role in driving T cell exhaustion, but the investigators do not provide strong evidence that this is truly happening in their animal model.

Reviewer #3 (Remarks to the Author):

As stated in my previous review the authors have done a series of experiments addressing all of my commentaries and have also addressed some of the most relevant comments from other reviewers. They also addressed my prior concern regarding blockade of PD-1/TIM-3 and TNF α . I believe this paper addresses an important issue in resistance to anti PD-1 with enough mechanistic insight to warrant publication at Nature Communications

Below is a point-by-point response in blue to Reviewer #2 comments in which we highlight why we do believe our manuscript warrants publication in Nature Communications:

1- The investigators show some data demonstrating the role of TNF in promoting a modest upregulation of Tim-3 by T cells in mouse and in TILs. In addition, using anti-PD1, anti-Tim-3, and anti-TNF blocking antibodies, the antitumor effects of dual PD1/TNF blockade appear superior.

In this study, the most striking result shows that combining anti-TNF and anti-PD-1 to treat established murine melanoma tumors induced rejection of 75% of tumors compared to only 30% of them with anti-PD1 alone. Moreover, we show in two additional murine tumor models (the 4T1 mammary tumor model and the Lewis Lung carcinoma model), which are intrinsically resistant to anti-PD-1 therapy, that blockade of TNF partially overcame this resistance to anti-PD-1. These findings highlight the **definitive superiority** of this combination treatment over the single anti-PD-1 therapy.

We also show that anti-PD-1 treatment significantly increased the percentage of TIM-3+ CD8+ T cells in the tumor microenvironment in a TNF-dependent manner. While we agree with Reviewer #2 that the proportion increase might be considered as "modest", the expression level of TIM-3 on CD8+ TIM-3+ TILs (as evaluated by mean fluorescence intensity) was however **substantially increased** (nearly doubled). It is well admitted that such increments may have significant biological impacts on T cells.

In these experiments, the role of Tim-3 in mediating some of the effects of TNF blockade is not demonstrated.

Although we agree as well with Reviewer #2 that additional mechanisms might be involved in promoting the anti-tumor immune response upon combination therapy, our results do show that CD8+ T cells expressing high levels of TIM-3 are more susceptible to AICD. In this context, TIM-3 could rather be considered as a marker of susceptibility of TILs to AICD, which would participate to dampen the response to anti-PD-1 therapy.

Investigation of any additional mechanism is out of the scope of the present study.

2- The experiments to evaluate TIL functions are made at the RNA level and upon stimulation with PMA/ionomycin. The investigators observed a slight decrease of IFN production by CD8+ TILs in the absence of TNF. They also report granzyme B expression by CD8+ TILs is in TNF-/- mice, independently of PD-1 blockade. These observations are of uncertain significance and do not explain the strong synergistic effects of PD-1 blockade and TNF depletion.

“The strong synergistic effects of PD-1 blockade and TNF depletion”, as acknowledged by Reviewer #2, are most likely due to the strong accumulation of CD8+ TILs in melanoma tumors from TNF-deficient mice treated with anti-PD-1 as depicted in Figure 2c. **CD8+ TILs are more than doubled in TNF-deficient mice as compared to wild-type mice upon anti-PD-1 therapy.** This phenomenon is associated with a significant reduction of activation-induced cell death (AICD) in CD8+ TILs in TNF-deficient mice (Fig. 2d).

As mentioned by Reviewer #2, production of IFN- γ as well as Granzyme B by CD8+ TILs was analysed by flow cytometry following PMA/ionomycin re-stimulation. This allowed us to evaluate the functional phenotype of TILs freshly extracted from murine melanoma tumors, which we believe is best reflecting their phenotype in the tumor microenvironment. In addition, we analysed the expression of PD-1, PD-L1, PD-L2, TIM-3, Ki67, LAG3, TIGIT, CTLA-4 as well the viability status of CD4+ and CD8+ T cells freshly extracted from tumors in the absence of PMA/Ionomycin re-stimulation.

We disagree with Reviewer #2' statement "**evaluations of TIL functions are made at the RNA level**". We did not evaluate the phenotype of TILs at the RNA level. RNA analyses were carried out to evaluate the expression of cytokines (especially IFN- γ and TNF) and chemokines on extracts from total tumors. Additionally, we used flow cytometry to determine whether TNF deficiency synergised with anti-PD-1 to modulate the production of IFN- γ by CD8+ TILs. The goal of these experiments was (i) to determine whether TNF deficiency enhanced the expression of IFN- γ upon PD-1

blockade in the tumor microenvironment and (ii) whether CD8+ TILs were directly or indirectly responsible for any fluctuation in IFN- γ levels in the tumor microenvironment.

On the one hand, upon anti-PD-1 treatment, TNF deficiency increased the level of transcripts coding for IFN- γ in the tumor microenvironment. Of note, this effect was abrogated in tumors grafted to CD8+ T cell-deficient mice, indicating that CD8+ TILs are required for potent IFN- γ production in melanoma tumors under our experimental conditions. On the other hand, flow cytometry analyses of CD8+ TILs showed that TNF deficiency induced a slight decrease in the proportion of IFN- γ -producing CD8+ T cells following anti-PD1 treatment. This slight decrease was obviously compensated by the strong accumulation of CD8+ TILs in TNF-deficient mice. Collectively, the data indicate that the global IFN- γ response was unlikely dampened but rather increased by TNF loss upon anti-PD-1 therapy. Moreover, lymphocytes from anti-PD-1-treated TNF-deficient mice likely remained cytotoxic as demonstrated by the conserved proportion of granzyme B+ CD8+ TILs in these animals compared to the WT ones (Supplementary Fig. 7d).

To sum up, as opposed to the statement "***These observations are of uncertain significance***", all these results show that the combination therapy does not affect the cytotoxic potential of CD8+ TILs, and even protects them from AICD.

Also, one wonders about the CD8+ T cell-mediated tumor reactivity, which is not tested in these experiments.

We did demonstrate in a previous study that TNF deficiency increased the proportion of CD8+ T cells reactive against the TRP-2 melanoma antigen (PMID: 25977337).

3- The authors provide new data showing less T cell death in CD8+ TILs isolated from TNF-/- mice treated with anti-PD-1 antibodies. The observation that PD-1 blockade would induce CD8+ TIL death goes against all the evidence that PD-1 regulates T cell survival and that PD-1 blockade increases T cell survival and function in the TME. They also show data suggesting that anti-PD-1 therapy triggered TNF-dependent AICD of PD-1+ Tim-3+ T cells.

We disagree with Reviewer #2 that “***PD-1 blockade would induce CD8+ TIL death goes against all the evidence that PD-1 regulates T cell survival and that PD-1 blockade increases T cell survival and function in the TME***”. Whereas the detrimental effect of the PD-1/PD-L1 signalling on T cell accumulation is indeed well documented in different pathological models, we observed an increased proportion of dead CD8+ TILs upon anti-PD1 treatment, an effect abrogated in the absence of TNF. This is rather expected taking into account that anti-PD-1 re-activate CD8+ TILs. Indeed, in various pathophysiological contexts, T cell activation is not only associated with cell proliferation but also cell death (i.e., activation-induced cell death, AICD). TNF was identified as being involved in AICD of CD8+ T cells by M. Lenardo’s group *in vitro* (PMID: 7566090).

This observation was restricted to CD8+ TILs expressing both PD-1 and TIM-3. TIM-3 ligation by its ligand is known to promote T cell death (PMID: 22863785; PMID: 18258591). As anti-PD-1 treatment not only increased the percentage of CD4+ and CD8+ TILs expressing TIM-3 but also the expression level of TIM-3 on CD8+ TIM-3+ TILs, we conclude that the beneficial impact of anti-PD-1 on T cell survival is likely counteracted by the overexpression of TIM-3 on CD8+ TILs.

This effect is very modest as % cell death within PD-1+Tim3+ is at next 7-8 % vs. 3-5% in Tim-3-PD-1-. It is very unlikely that such small increase in % dead cells is responsible for the clinical effects of dual TNF/PD-1 blockade.

The proportion of PD-1+ TIM-3+ TILs undergoing AICD following anti-PD-1 therapy was significantly increased (nearly doubled) at day 10 in wild-type mice as compared to TNF-deficient mice. This will likely have a substantial clinical impact over time. Indeed, several studies demonstrated that such "modest" levels of T cell death could have a significant impact on immune responses in other *in vivo* pathophysiological contexts (PMID: 28538737; PMID: 23170254).

Overall, the mechanisms of action of the dual TNF/PD-1 blockade remain uncertain. As suggested by Beyer et al., TNF may play a critical role in driving T cell exhaustion, but the investigators do not provide strong evidence that this is truly happening in their animal model.

It is well established that TIM-3 expression on T cells is associated with the occurrence of an exhausted phenotype (as mentioned by Reviewer #2). The work by Beyer et al. quoted by Reviewer #2 deals with a totally different system, that is TNF-induced exhaustion of CD4+ cells in a model of chronic viral infection. In our model, the kinetics of tumor regression in our melanoma model does not allow for exhaustion to take place. As stated in response to comment #1, TIM-3 in this context is rather associated with an increased susceptibility to AICD. It is however likely that exhaustion will occur at later time points for TIM-3+ TILs, which resist AICD in wild-type mice upon anti-PD-1 therapy. Evaluating such exhaustion process in wild-type mice would not add essential novel information as regard to the state of the art. This is thus not a critical issue to support our main conclusion.